# Neurocognitive analyses reveal that video game players exhibit enhanced implicit temporal processing

Francois R. Foerster [1✉], Matthieu Chidharom[1,2], Anne Bonnefond[1] & Anne Giersch [1]

Winning in action video games requires to predict timed events in order to react fast enough. In these games, repeated waiting for enemies may help to develop implicit (incidental) preparation mechanisms. We compared action video game players and non-video game players in a reaction time task involving both implicit time preparations and explicit (conscious) temporal attention cues. Participants were immersed in virtual reality and instructed to respond to a visual target appearing at variable delays after a warning signal. In half of the trials, an explicit cue indicated when the target would occur after the warning signal. Behavioral, oculomotor and EEG data consistently indicate that, compared with non-video game players, video game players better prepare in time using implicit mechanisms. This sheds light on the neglected role of implicit timing and related electrophysiological mechanisms in gaming research. The results further suggest that game-based interventions may help remediate implicit timing disorders found in psychiatric populations.

[1] Université de Strasbourg, INSERM U1114, Pôle de Psychiatrie, Centre Hospitalier Régional Universitaire de Strasbourg, Strasbourg, France. [2] Department of Psychology, Lehigh University, Bethlehem, PA, USA. ✉email: francoisfoerster@gmail.com

Video games are widely accessed and consumed at all ages. Within a few decades, research showed that playing video games can enhance cognition[1–3]. This enhancement involves increased ability to learn on the fly[4–6] and improved attentional control[7,8]. Game-induced cognitive enhancements depend on the gameplay. The extent of these enhancements remains unclear[9]. In action video games, such as first-person shooter games, a key component is the ability to predict in time the appearance of visual targets, often foes, to prevail. In this case, being 'on time' involves both implicit and explicit time predictions[10–13]. Explicit timing refers to any task in which participants receive explicit instruction to process temporal information. In contrast, during implicit timing tasks, participants are unaware of processing time. Timing intervenes in many tasks incidentally. Firstly, action video game play requires motor responses that are bound to temporal preparation mechanisms[14,15]. Secondly, several stimuli occurrences are expected during the games. Indeed, in most first-person shooter games subjects expect and wait for enemies to appear. Even though the time of occurrence of the enemy is stochastic, the probability of an enemy occurrence is high, and gamers can prepare to react as fast as possible when the enemy finally appears. Players likely benefit from practice to refine temporal expectations of forthcoming targets, even if they are unaware of such expectations. In turn, expectations might optimize preparation and speed up reactions. Therefore, the intensive training of action video game players might improve implicit (incidental) temporal mechanisms. In addition, players might use explicit temporal cues in games. For instance, they might explicitly use a visual cue to predict the exact moment of targets in their visual field, i.e., consciously orienting their attention in time[8]. A large number of studies revealed that playing action video games improves explicit spatial attention mechanisms, their evolution over time and development, or explicit temporal abilities[8,16–23]. However, it is still unclear whether VGPs learn to implicitly benefit from the passage of time to better anticipate future targets. We need to know which types of mechanisms are enhanced in video game players, to better understand the impact of video game, and how they may help pathological groups. For example, implicit temporal mechanisms appear to be impaired in patients with schizophrenia (see ref. [24] for a review), and video games have been proposed as a potential rehabilitation tool for psychiatric disorders such as schizophrenia[25]. Knowing how video game play shapes brain mechanisms and behaviors will help to adapt these rehabilitation tools to pathologies.

In this cross-sectional study, we investigated implicit and explicit prediction and preparation in time in action video game players (VGPs) and non-video game players (NVGPs) using a variable foreperiod task in a virtual environment. In this task, participants had to anticipate to speed up their responses to a visual target. Virtual reality helps to get close to ecological conditions, enables an optimal commitment to the task and allows the gaze to be tracked in the 3D space using the embedded eye-tracking system of the headset. We tested whether reaction time performance is enhanced in VGPs and relies on enhanced implicit processing of the passage of time (hypothesis 1), enhanced explicit orientation of attention in time (also called temporal orienting; hypothesis 2), or both (hypothesis 3). EEG and oculomotor activities were concomitantly recorded to evaluate the neurocognitive mechanisms responsible for these potential enhancements.

In our task, a target occurs at varying delays after an initial warning signal, hence the name of variable foreperiod (FP) task[26]. Participants were made aware of the two possible foreperiods (also called inter-stimulus intervals, ITIs), i.e., 400 ms (short FP) or 1000 ms (long FP). The warning signal and target were embedded in robots, which created an environment closer to

video games and more entertaining than traditional computer-based tasks. Participants reacted to the target by pressing a button as fast as possible. The probability of target occurrence increases with the elapsing time (50% at 400 ms and 100% at 1000 ms) and participants benefit from the passage of time to prepare their response, leading to faster reaction times in long than short FP[26–30]. This preparation indexes the implicit processing of the passage of time (neutral cue condition). In contrast, in temporal orienting a cue (in our case a robot's color, Fig. 1; temporal cue condition) explicitly indicates the foreperiod: the participant is trained to use and associate the cue with the timing of the target occurrence before the task. The cue orients attention in time and yields a decrease in reaction time.

Several neurobiological indexes are associated with temporal processing. The EEG contingent negative variation (CNV) is a neuronal signal known to be increased during temporal orienting[31–37], whereas theta-band oscillations have been found to increase when a visual target is implicitly expected[38]. Finally, temporal orienting has been associated with small fixational saccades, called microsaccades, which are inhibited before the onset of a temporally predictable sensory signal[39–42]. Here, all evidence shows enhanced implicit temporal processing in VGPs.

## Results

**Reaction times.** VGPs are believed to be impulsive (but see ref. [43]) amongst the general public. A preliminary analysis of premature responses (anticipation errors—responding before the target appearance) revealed no evidence of increased impulsivity in VGPs compared with NVGPs (see Supplementary Note 1). Then, a three-way rANOVA with the factors Group, Cue and Foreperiod was performed on reaction times (Fig. 2). No effect of the Group was revealed ($p = 0.162$). However, significant main effects of the Cue (Mean$_{Temporal}$ = 346 ms, CI$_{Temporal}$ = 11.1 ms; Mean$_{Neutral}$ = 355 ms, CI$_{Neutral}$ = 9.6 ms; F(1, 44) = 11.9, $p = 0.001$; $\eta^2_p = 0.214$) and the Foreperiod (Mean$_{ShortFP}$ = 359 ms, CI$_{ShortFP}$ = 9.8 ms; Mean$_{LongFP}$ = 342 ms, CI$_{LongFP}$ = 10.7 ms; F(1, 44) = 70.8, $p < 0.0001$; $\eta^2_p = 0.617$) were revealed. These effects indicated that participants were faster to respond when the foreperiod was long rather than short (i.e., implicit processing) and when the timing of the target was predictable (i.e., explicit processing). The rANOVA revealed an interaction effect between the Cue and the Foreperiod (F(1, 44) = 22.5, $p < 0.0001$; $\eta^2_p = 0.339$). Planned comparisons showed that the effect of the Foreperiod was absent in the temporal cue condition ($t(45) = 12.6$, $p > 0.11$). All these results replicate those in the literature, showing that they are preserved in the virtual environment. Finally, the analysis also revealed a triple interaction between the Cue, the Foreperiod and the Group (F(1, 44) = 4.89, $p = 0.032$; $\eta^2_p = 0.100$), with an effect of the Foreperiod in the neutral cue condition in VGPs ($t(45) = 25$, $p = 0.003$) but not in NVGPs ($t(45) = 17.3$, $p = 0.12$). Consequently, only VGPs benefited from the passage of time when the target was not temporally cued—they were faster when the foreperiod was long rather than short. According to the learning to learn theory of game-induced cognitive enhancements[4,5], VGPs might have learned to optimize task performance more rapidly than NVGPs. Results from additional frequentist and Bayesian analyses do not support this claim (see Supplementary Note 2).

Also, the lack of effect in NVGPs might have been due to a larger inter-individual variability in RTs in comparison with VGPs (see Supplementary Fig. 1). We thus propose a different approach to estimate the subject-wise benefits from both the implicit passage of time and explicit temporal orienting to the task performance.

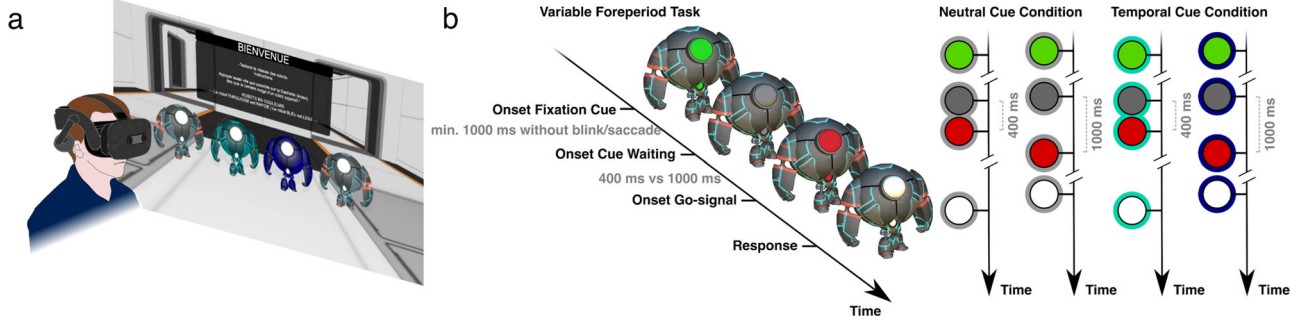

**Fig. 1 Virtual reality setup and experimental task. a** Action video game players (VGPs, N = 23) and non-video game players (NVGPs, N = 23) were immersed in a virtual environment with robots and performed a variable foreperiod task. **b** A short (400 ms) or long (1000 ms) foreperiod separated in time the offset of an initial warning signal (i.e., a green light) and the onset of a target (i.e., a red light). The task of the participants was to press a button as fast as possible after the appearance of the target. The color of the robots served as a predictive cue (blue and turquoise robots; temporal cue condition) or not (gray robots; neutral cue condition) for the timing of the target. Reaction times, eye-tracking, and EEG measures were used to evaluate whether or not, and how, VGPs and NVGPs benefited from implicit temporal expectations and explicit temporal attention to optimize their performance.

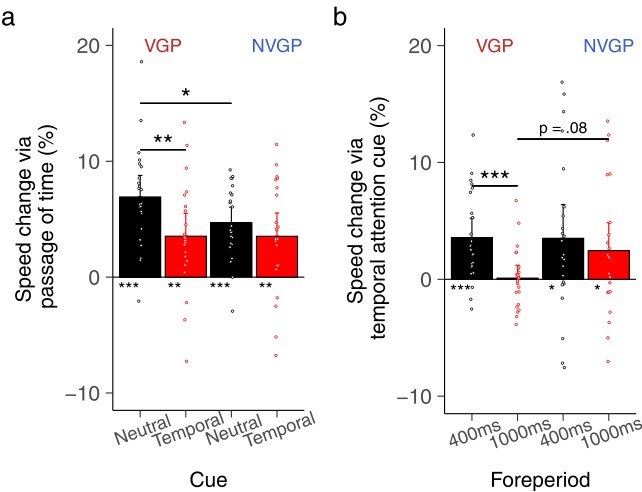

**Fig. 2 Behavioral results. a** Indexes were calculated to evaluate the effect of the implicit passage of time and explicit temporal attention on reaction times while accounting for inter-individual variability. To assess the benefit from the passage of time provided by a longer FP, an index was calculated as follows: Speed Change (%) = 100 * (RT$_{short\ FP}$ − RT$_{long\ FP}$/RT$_{short\ FP}$).
**b** Similarly, to evaluate the benefit from temporal attention provided by the temporal cue, another index was calculated as follows: Speed Change (%) = 100 * (RT$_{Neutral}$ − RT$_{Temporal}$/RT$_{Neutral}$). Error bars represent ±one confidence interval of the mean. *$p$ < 0.5, **$p$ < 0.01, ***$p$ < 0.001. VGPs (N = 23) benefited more than NVGPs (N = 23) from the implicit passage of time.

### Estimates of the passage of time and temporal orienting effects.

According to the literature, participants should respond faster when the foreperiod is long rather than short (RT$_{LongFP}$ < RT$_{ShortFP}$) and when the foreperiod is predicted by the cue (RT$_{Temporal}$ < RT$_{Neutral}$). These effects represent the benefit provided by the passage of time and by temporal orienting, respectively. To evaluate these benefits, we computed subject-wise indexes that consider the between-subject variability of response times.

First, we quantified how much participants implicitly benefited from the passage of time. We calculated the percentage of speed change given the formula:

$$Speed\ change(\%) = 100 * (RT_{shortFP} - RT_{longFP}/RT_{shortFP}) \quad (1)$$

The RT$_{short\ FP}$ was used as denominator to evaluate how faster a participant can be in a specific condition (long FP) relative to a

baseline condition (short FP). A control analysis revealed similar observations when considering the sum of the two conditions as denominators. A one-sample $t$-test analysis conducted on the percentage of speed change revealed that all participants, independently of their group, benefited from the passage of time in both the neutral (all $p$ < 0.0001) and temporal (all $p$ < .0015) cue conditions. Given such benefit was not observed in the typical comparisons of RT$_{short\ FP}$ vs. RT$_{long\ FP}$ these results suggest our calculation is best suited to evidence the benefits from the passage of time. We then performed a three-way rANOVA with the factors Cue, Block, and Group to assess whether VGPs took better advantage of the passage of time than NVGPs across the cueing conditions and the duration of the experiment. The rANOVA showed a main effect of the Cue (F(1, 44) = 19.77, $p$ < 0.0001; $\eta^2_p$ = 0.31) and an interaction effect between the Cue and the Group (F(1, 44) = 4.7, $p$ = 0.036; $\eta^2_p$ = 0.097). VGPs benefited from the passage of time significantly more in the neutral cue condition (Mean$_{Neutral}$ = 6.92%, CI$_{Neutral}$ = 1.86%) than in the temporal cue condition (Mean$_{Temporal}$ = 3.53%, CI$_{Temporal}$ = 1.98%, $t(45)$ = 3.39, $p$ = 0.0013). This effect was absent in NVGPs (Mean$_{Neutral}$ = 4.69%, CI$_{Neutral}$ = 1.35%, Mean$_{Temporal}$ = 3.53%, CI$_{Temporal}$ = 2.01%, $t(45)$ = 1.17, $p$ = 0.25). Crucially, planned comparisons showed that the benefit of the passage of time was greater in VGPs than in NVGPs in the neutral cue condition ($t(45)$ = 2.23, $p$ = 0.013) but not with the temporal cue condition ($t(45)$ = 0.007, $p$ = 0.995). No main effect ($p$ = 0.334) nor interaction effects with the factor Block (all $p$ > 0.332) was reported, as further illustrated in Supplementary Fig. 1c.

Second, we quantified how much participants explicitly benefited from the temporal cue to speed up their response time for each foreperiod. We calculated the percentage of speed change given the formula:

$$Speed\ change(\%) = 100 * (RT_{Neutral} - RT_{Temporal}/RT_{Neutral}) \quad (2)$$

To evaluate whether VGPs took better advantage of the temporal cue than NVGPs, a three-way rANOVA was performed on these values with the factors Foreperiod, Block, and Group. The analysis revealed a main effect of the Foreperiod (F(1, 44) = 20.19, $p$ < 0.0001; $\eta^2_p$ = 0.315) and an interaction effect between the Foreperiod and the Group (F(1, 44) = 5.75, $p$ = 0.021; $\eta^2_p$ = 0.116). VGPs benefited from the temporal cue at short FP ($t(22)$ = 4.47, $p$ = 0.0002) but not at long FP ($t(22)$ = 0.19, $p$ = 0.85), resulting in a significant difference in the effect of the temporal cue between the two foreperiods ($t(45)$ = 3.47, $p$ = 0.0007). NVGPs benefited from the temporal cue at both foreperiods (all $p$ < .044), hence there was

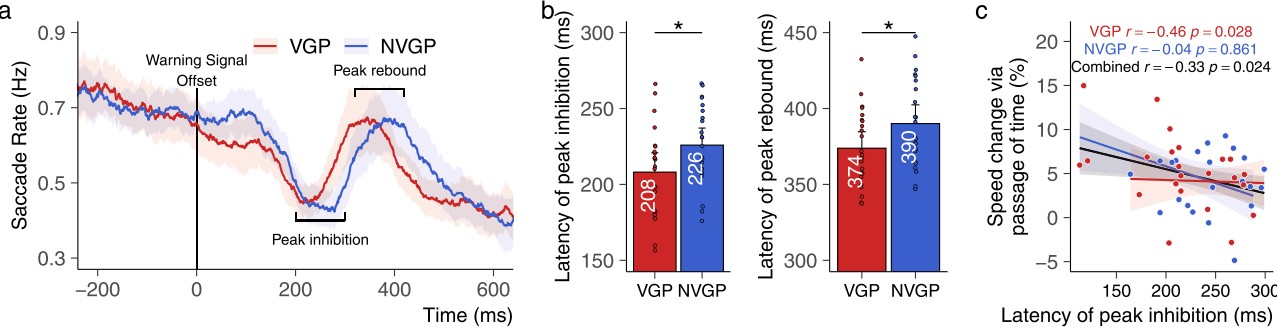

**Fig. 3 Dynamics of saccade rates. a** We observe an oculomotor reflex when the foreperiod starts. **b** This oculomotor reflex was faster in VGPs ($N = 23$) than NVGPs ($N = 23$). **c** The negative correlation between the benefit from the implicit passage of time and the latency of peak inhibition shows that VGPs' benefitting from the implicit passage of time had faster oculomotor reflexes. *$p < 0.5$. Colored shaded areas represent ±one SEM. Error bars represent ±one confidence interval of the mean.

no difference in the effect of the temporal cue between the two foreperiods ($t(45) = 1.05$, $p = 0.53$).

A control analysis revealed no correlation between the education level and the benefits from the passage of time, nor between the education level and the benefit from the temporal cue. Complementary results on the effect of the FP at trial $t_{-1}$ on the reaction time at trial $t$ can be found in Supplementary Note 3 and are depicted in Supplementary Fig. 2.

**Faster oculomotor responses in action video game players**. We investigated the peak of reflexive saccadic inhibition and rebound[44–47] evoked by the offset of the initial warning signal representing the start of the waiting period. In other tasks, the saccadic inhibition represented the enhanced stimulus processing resulting from the top-down influence of attention[48,49].

The latencies of the peak of saccadic inhibition and rebound are in accordance with previous studies that used more conventional eye-tracking systems[45,48]. To evaluate the oculomotor responses to the start of the waiting period, a three-way rANOVA with the factors Group, Foreperiod, and Cue was conducted on both the latencies of the peak of inhibition and the latencies of the peak rebound extracted within the 100-300 ms and 300-500 ms time-windows, respectively. The analysis of the peak of inhibition revealed a main effect of the Group (F(1, 44) = 4.7, $p = 0.035$; $\eta^2_p = 0.097$) and indicated that the peak of inhibition occurred significantly earlier in VGPs (mean = 208 ms, CI = 12 ms) than in NVGPs (Mean = 226 ms, CI = 11 ms). No other effect was found (all F(1,44) < 1.4, all $p > 0.243$). Similarly, the Group factor influenced the latencies of the peak rebound (F(1, 44) = 4.18, $p = 0.047$; $\eta^2_p = 0.087$): the peak occurred earlier in VGPs (Mean = 374 ms, CI = 11 ms) than NVGPs (Mean = 390 ms, CI = 12 ms). An interaction effect between the Group and the Cue (F(1, 44) = 5.18, $p = 0.043$; $\eta^2_p = 0.105$) indicated that the earlier peak rebound in VGPs relative to NVGPs was specific to the temporal cue condition ($t(45) = 35.2$, $p = 0.001$). Overall, the analysis suggests a faster oculomotor response in VGPs (Fig. 3a–c). Interestingly, a Pearson correlation revealed a significant negative correlation between the latency of the peak inhibition and the benefit from the passage of time ($N = 23$, $r = -0.46$, $p = 0.028$) in VGPs but not in NVGPs ($N = 23$, $r = -0.04$, $p = 0.861$), independently of the cueing condition. In this analysis, all data points were within the interval of ±3 SD from the mean. Thus, no outliers were detected and all participants were included in the correlation analysis. This suggests that the benefit from the passage of time was more important in participants with short latency of peak inhibition (Fig. 3c), which is particularly the case in VGPs. Finally, the temporal orienting phenomenon has been associated with small saccades during gaze fixation, which is inhibited before the onset of a temporally predictable target[39–42]. We replicated such results with our paradigm (see Supplementary Note 4 and Supplementary Fig. 3). There was no significant group effect on the amplitude of saccade rates at peaks.

*CNV and temporal orienting.* Several neurobiological markers are associated with temporal expectations. The contingent negative variation (CNV) is an EEG signal whose magnitude is increased during temporal orienting[31–37]. A two-way rANOVA with the factors Group and Cue was conducted on the magnitude of the centro-parietal CNV (Fig. 4a) recorded within the 400-1000 ms time interval in trials with long FP only (i.e., when the target appeared at 1000 ms). This time interval starts at the time of the earliest possible occurrence of the target (400 ms) and ends when the target appears (1000 ms). The analysis did not report a significant effect of the Group (F(1, 43) = 3.4, $p = 0.071$; $\eta^2_p = 0.074$), but revealed a main effect of the Cue (F(1, 43) = 7.2, $p = 0.01$; $\eta^2_p = 0.143$; Fig. 4b-c; see Supplementary Note 5 and Supplementary Fig. 4) indicating that the magnitude of the CNV was larger in the neutral cue condition (mean = −5.48, CI = 1.35) than in the temporal cue condition (mean = −3.99, CI = 1.33, Fig. 4d). No interaction effect (F(1, 43) = 0.03, $p = 0.88$; $\eta^2_p < 0.001$) was revealed. Hence, at first sight, this result seems to contradict the literature.

Previous studies showed that the CNV slope is adjusted according to the temporal expectations, in a way that the magnitude of the CNV reaches its maximum around the expected time of a target appearance[32,33,39]. We evaluated the slope in trials with the long FP only, calculated as the difference between the averaged CNV in the 900-1000 ms time interval and the averaged CNV in the 300–400 ms time interval (at which time there was no target since we considered trials with the long FP only), divided by the time difference between the two windows (i.e., 0.6 s). Here, a flat or positive slope would indicate that the CNV peaked (i.e., was more negative) around the short FP, whereas a negative slope would suggest that the CNV peaked around the long FP. One-sample $t$-tests indicated the presence of a negative slope across all cues and groups (all $t(21, 22) > 2.24$, all $p < 0.036$, Fig. 4e) except in the temporal cue condition in VGPs ($t(21) = 0.93$, $p = 0.36$). According to the literature, this result suggests that VGPs were equally prepared at short and long delays when they knew that the target would appear at 1000 ms. The Group or the Cue did not affect the steepness of the slopes (all F(1, 43) < 1.91, all $p > 0.175$). However, a Pearson correlation analysis revealed that participants who benefited from the temporal cue in trials with long FP had a more negative CNV slope in trials with the temporal cue rather than with the neutral

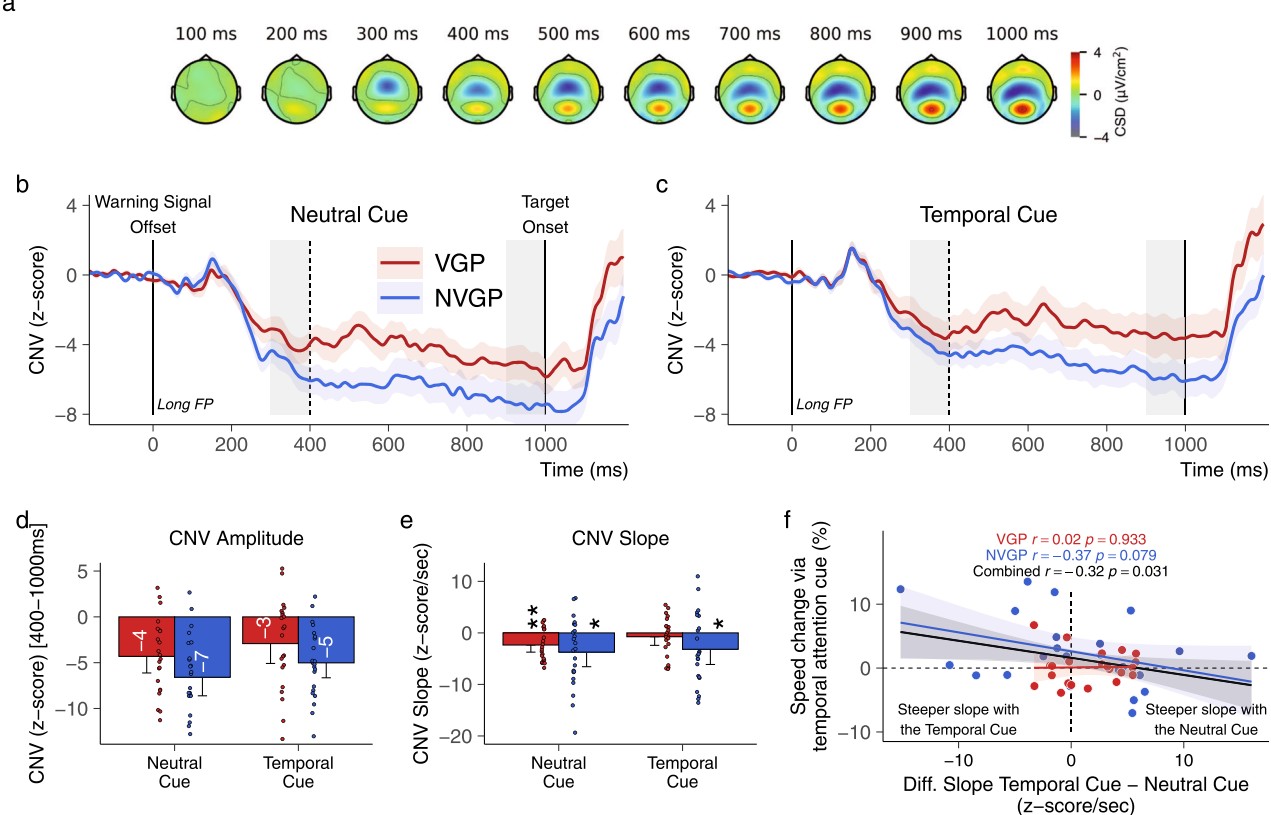

**Fig. 4 Contingent negative variation. a** Temporo-spatial evolution of the normalized CNV in trials with the long foreperiod across cueing conditions. In this analysis, we evaluated the explicit temporal orienting given (1) the averaged amplitude of the CNV within the 400-1000 ms time interval and (2) the slope of the CNV representing the amplitude difference between 1000 ms and 400 ms. In trials with the neutral (**b**) and temporal (**c**) cues, the amplitude of the CNV was not significantly reduced in VGPs ($N = 22$) relative to NVGPs ($N = 23$). Gray areas represent the time intervals used to calculate the slope of the CNV. **d** Independently of the group, the amplitude of the CNV was reduced in the temporal cue condition relative to the neutral cue condition. **e** All CNV slopes were negative, except in VGPs in the temporal cue condition. **f** Correlation analysis indicated that participants benefiting from the temporal cue in trials with the long FP exhibited a more negative CNV slope in the temporal rather than in the neutral cue condition. Colored shaded areas represent ±one SEM. Error bars represent ±one confidence interval of the mean. *$p < 0.5$, **$p < 0.01$.

cue ($N = 46$, $r = -0.323$, $p = 0.031$, Fig. 4f). This correlation was not group-specific. This result supports the literature suggesting that the CNV slope reflects the explicit temporal orienting phenomenon[39].

**Reduced theta oscillations in action video game players**. In the literature, temporal expectations were reflected in the centrally recorded theta-band power and the centro-motor theta and beta phase-amplitude coupling (PAC; see Supplementary Fig. 5)[38]. A two-way rANOVA with the factors Group and Cue was conducted on the averaged theta-band power in the 300-500 ms time interval (in the same time interval as in ref. [38]), for trials with a long FP (Fig. 5a–c). This time interval of 300-500 ms corresponds to the short foreperiod delay. During this time interval, the probability of target occurrence is 50% and 0% in the neutral and temporal cue conditions, respectively. Hence, at 300-500 ms in trials with a long foreperiod, target expectation is stronger in the neutral than in the temporal cue condition. The analysis reported a main effect of the Cue ($F(1, 43) = 6.05$, $p = 0.018$; $\eta^2_p = 0.123$), revealing that the magnitude of theta oscillations was increased in the neutral (Mean = 12.10, CI = 2.65) relative to the temporal cue condition (Mean = 8.86, CI = 2.88). Also, the analysis revealed a main effect of the Group ($F(1, 43) = 6.49$, $p = 0.014$; $\eta^2_p = 0.131$), with a reduced magnitude of the theta oscillations in VGPs (Mean = 7.53, CI = 2.08) compared with NVGPs (Mean = 13.31, CI = 3.11, Fig. 5d). No interaction effect was reported ($F(1, 43) = 1.07$,

$p = 0.307$). These results suggest that (1) theta-band activity was increased when the target probability occurrence was higher (i.e., in the neutral cue condition), thus supporting the link between temporal expectation and mid-frontal theta-band activities, and (2) VGPs had decreased temporal expectations relative to NVGPs when the probability of target occurrence was indeed low (i.e., 50 or 0%).

**Increased phase-amplitude coupling (PAC) in action video game players**. The interplay of multiple brain rhythms permits efficient communication between distant cortical areas. To assess this communication, comodulograms were calculated on the 0–1000 ms time interval for trials with a long FP. Their visual inspection revealed a coupling between the phase of the fronto-central theta oscillations and the amplitude of the left motor beta oscillations (20–40 Hz range; Fig. 6a–c). Given the non-normality of the θ-β PAC values (Shapiro-Wilk tests $p < 0.001$), non-parametric two-sided Wilcoxon signed-rank tests were used to evaluate significant differences across groups and cueing conditions. The PAC was evident in all groups and cueing conditions (all $p < 0.0001$; one-sample Wilcoxon signed-rank tests). The analysis revealed no main effect of the Group ($p = 0.478$) or the Cue ($p = 0.087$). The θ-β PAC values were not statistically different between groups in trials with the neutral ($p = 0.067$) or temporal ($p = 0.71$) cue conditions. However, in VGPs these θ-β PAC values were significantly higher in the neutral cue condition

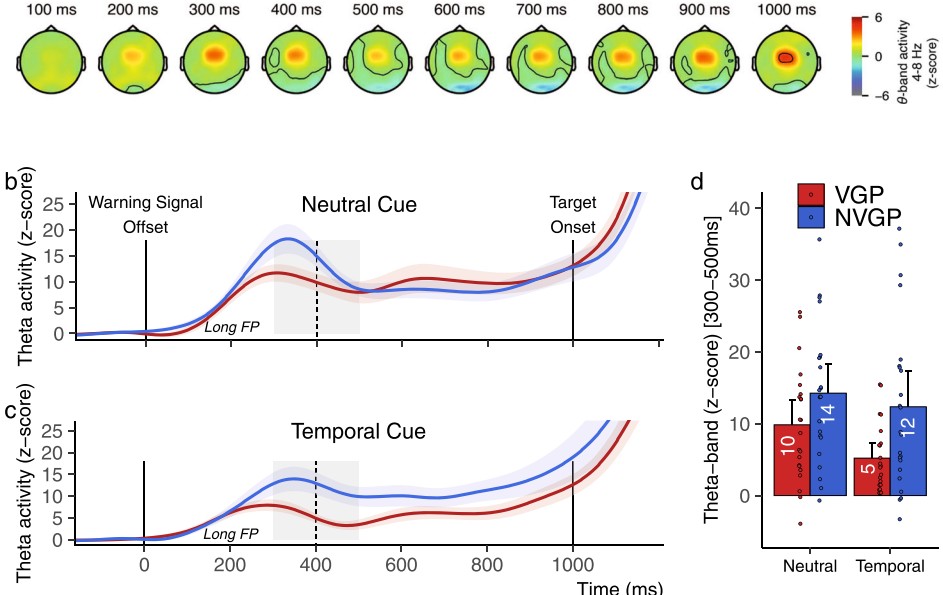

**Fig. 5 Mid-frontal theta-band oscillations. a** Temporo-spatial evolution of the theta-band power in trials with the long foreperiod across cueing conditions. Theta-band power increased around the time interval of the first possible foreperiod in both neutral (**b**) and temporal cue (**c**) conditions. Gray areas represent the time interval used for the analysis. Within this interval, the probability of target occurrence is 50% and 0% in the neutral and temporal cue conditions, respectively. Colored shaded areas represent one standard error of the mean. **d** Theta-band power is increased when the target expectation is stronger, that is in the neutral cue condition. Also, theta-band power is reduced in VGPs ($N = 22$) suggesting that VGPs have decreased temporal expectations relative to NVGPs ($N = 23$) when the probability of target occurrence is low (i.e., 50 or 0%). Error bars represent ±one confidence interval of the mean.

(Mean $= 9.3 \times 10^{-5}$, CI $= 3.9 \times 10^{-5}$) relative to the temporal cue condition (Mean $= 6.8 \times 10^{-5}$, CI $= 4.2 \times 10^{-5}$; $p = 0.039$, moderate size effect $r = 0.31$). This effect was absent in NVGPs ($p = 0.71$). A Spearman correlation indicated a strong relationship between the log-transformed θ–β PAC values and the benefit from the implicit passage of time in the neutral cue condition (all $p < 0.013$, Fig. 6d) but not in the temporal cue condition (Fig. 6e). These results expand previous findings[38,50,51] but also suggest that participants benefiting from the implicit passage of time exhibit an increased fronto-motor functional oscillatory connectivity, which is especially the case in VGPs. Multiple control analyses strongly support the specificity of the θ-β PAC in the frequency- and space-domains (see Supplementary Note 6).

## Discussion

In this study, both implicit processing of the passage of time and explicit temporal attention were investigated in VGPs and NVGPs using a visual variable foreperiod task. Two foreperiods defined two possible time intervals separating the offset of an initial warning signal (i.e., a green light) and the onset of a target (i.e., a red light). These two foreperiods were known to the participants. Small fixation saccades and EEG of participants were monitored while they were instructed to provide a fast-manual response to the target. In this paradigm, the timing of the target appearance was predictable based on (1) the implicit passage of time given the conditional probability that the target has not appeared yet and (2) the explicit temporal orienting of attention given the cue. The decrease in reaction times when the foreperiod was long rather than short indicated that both VGPs and NVGPs anticipated the target in time. Regarding our hypotheses, implicit rather than explicit temporal skills were improved in VGPs. VGPs benefited more from the implicit passage of time than

NVGPs. We found three mechanisms related to this implicit passage of time benefit. First, the reflexive saccadic response to the offset of the warning signal occurred earlier in VGPs. This saccadic response predicted how much VGPs benefited from the passage of time, hence suggesting a relationship between the detection of the onset of a time interval of interest and the ability to track the elapsing time within this interval in VGPs. Second, the midline frontal theta-band activity was reduced in VGPs when the probability of occurrence of the target was low, suggesting an adequate adaptation of expectation to the probability of target occurrence. Third, the EEG analysis revealed a fronto-motor phase-amplitude coupling during the foreperiod, supporting previous interpretations of this coupling as a mechanism of implicit temporal processing[38]. Separate analyses in each group confirmed an increased fronto-motor phase-amplitude coupling in VGPs while performing the task in the neutral cue condition. Overall, the results suggest that VGPs have optimized implicit temporal skills allowing them to deploy and withhold cognitive resources when suitable.

The evidence suggests enhanced automatic mechanisms in video game players allowing them to time their perception even when they do not have to think about time itself. Indeed, the goal of the participant was to react to the target and not to time the foreperiods. Enhanced implicit mechanisms in VGPs start with faster saccadic reflexes, indicating an improved processing of the offset of the warning signal. The correlation linking reflexive oculomotor response to (at least some aspect of) the processing of the elapsing time appears in line with the hypothesis that saccades modulate accumulation processes in the brain[52]. In our task, tracking the elapsing time can be understood as a continuous accumulation of sensory evidence up to the target appearance. We speculate that faster saccadic reflexes permit to free cognitive

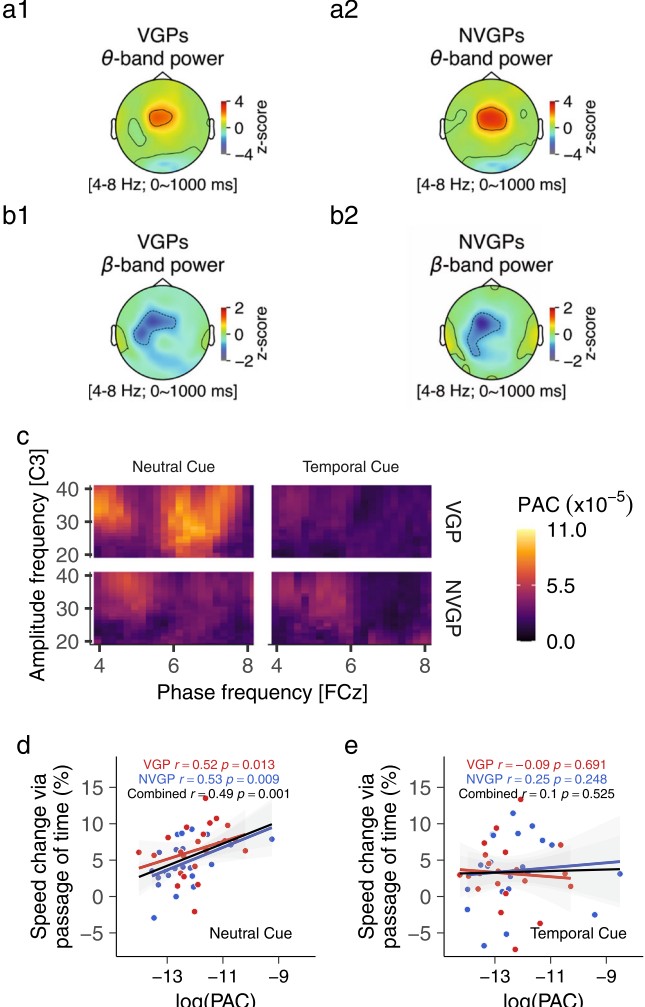

**Fig. 6 Fronto-motor phase-amplitude coupling.** Oscillations in theta (**a1–2**) and beta (**b1–2**) frequency bands were localized around fronto-central and left motor areas, respectively. **c** Comodulograms revealed a stronger θ–β phase-amplitude coupling (PAC) in the neutral cue condition than in the temporal cue condition. This effect was specific to the VGPs (N = 22). A Spearman correlation revealed a positive relationship between the participants' θ–β PAC (log-transformed values) and their benefit from the implicit passage of time (all participants). The correlation was significant with trials from the neutral cue condition (**d**) but not with trials from the temporal cue condition (**e**). Colored shaded areas represent ±one SEM.

resources, which in turn allows for more efficient accumulation processes[52]. Next, the theta-band analysis revealed that temporal expectations were lower in VGPs than in NVGPs when the probability of occurrence of the target was indeed low (50 or 0% chance). This suggests that, generally, temporal expectations were better adapted in VGPs. This better adaptation could have allowed VGPs to deploy distinct mechanisms to perform the task in our two conditions. It would explain the stronger θ–β functional coupling in VGPs when dealing with the uninformative neutral cue relative to the temporal cue. This coupling could represent a sensorimotor updating mechanism[53] integrating the elapsing time to refine implicit expectations about the timing of the target. All things considered, implicit rather than explicit temporal mechanisms appear optimized in VGPs in our study.

In this variable foreperiod task, it is in the neutral cue condition that VGPs differed from NVGPs the most (hypothesis 1). It is not surprising given the importance of implicit temporal

expectations to speed up responses in action video games, where some events have to be expected to optimize fast responses. Given the diverse evidence of improved attention in VGPs, enhanced benefit from the temporal cue might have been expected in these participants (hypothesis 2) but no proof was unveiled. Nevertheless, our data questions the EEG marker of temporal attention, namely the CNV. While we found increased CNV amplitudes when the timing of the target was neutrally cued, previous studies have reported increased CNV amplitudes when the timing of the target was temporally cued[35,36]. Crucially, in these studies the processing of the temporal cue was concomitant to the start of the waiting period, meaning an overlap of the encoding and usage of the temporal information and the continuous processing of the elapsing time. The paradigm presented here allows disentangling these two cognitive processes in assessing specifically whether and how the encoded temporal information helps to orient attention in time. With this important methodological distinction in mind, the pattern of CNV activity suggests that temporal attention reduces the neural cost of motor preparation, at least once the cued information has been processed. This result is consistent with fMRI data showing that temporal cue involves less activation in the right inferior frontal gyrus than neutral cue at long fore-periods, i.e., when updating is required[54].

Current theories suggest that action video game play increases neural plasticity[2], which in turn facilitates the rapid learning of the critical aspect of the task at hand[4] and explains the perceptual and cognitive enhancements found in VGPs. Similarly, we believe that the benefit from the passage of time found in VGPs may relate to transfer learning mechanisms[55]. Such transfer learning mechanisms explain why playing specific video games can speed up phonological decoding[19], enhance reading[56] and multitasking abilities[18]. In most first-person shooter games, such as Call of duty, Counter Strike or Unreal Tournament series, temporal sequences require the player to expect successive events, track the time elapsing between these events, and prepare to allocate their attention in space and time accordingly. Here, we propose that being trained to expect and attend to successive events accurately in time helps to learn to benefit from the passage of time in general.

Unfortunately, the present study cannot draw strong causal inferences on the effect of game play on temporal cognition. We deem that further work should evaluate the causal effect of action video game play on the implicit processing of time using a longitudinal approach. Also, future research should consider the possible influence of prior experience of VR, while keeping in mind that VR interventions could represent an affordable and engaging remediation tool for time perception alterations in psychiatric populations[25].

## Methods
**Participants.** In this cross-sectional study, participants were chosen using overt recruitment and screening criteria but were not aware of the specific aims of the study. The VGP group concerned 23 participants (four females, two left-handed, age mean = 25.2; SD = 5.7). The criterion to be considered a VGP was a minimum of 5 h per week of action video game practice for the previous 12 months, as reported in previous studies[4,8,22]. The games mainly included first-person shooters (e.g., Call of Duty series, Apex Legends, Overwatch, Counter Striker series), multiplayer online battle arena (e.g., Leagues of Legends, Heroes of the Storm), and real-time strategy (Starcraft II) which involve important visual and timing expectations, as well as high-speed visual processing and motor responses to optimize game performance. The NVGP group concerned 23 participants (seven females, five left-handed, age mean = 26.8; SD = 4.6). The criterion to be included in the NVGP group was little or no action video game practice for a minimum of 1 year, although no extensive practice ever (N = 16) was highly favored. T-test revealed no difference in age (p = 0.312) and Chi$^2$ -test revealed no difference in gender (p = 0.391) between VGPs and NVGPs. However, the education level was slightly lower in VGPs (mean = 14.3 years) than NVGPs (mean = 15.7 years, p = 0.012). All subjects had normal or corrected-to-normal visual acuity, as checked with the Freiburg Visual Acuity Test[57] (estimates of visual acuity did not vary between groups, p = 0.329). The exposure to VR experience was not quantified nor

compared between the two groups. No participant reported any feeling of cyber-sickness during or after the experiment. One VGP has been removed from the EEG analysis due to excessive noise in the recorded signal. All participants were given a compensation of 45€ for their participation and provided written informed consent to take part in the study. The study has been approved by the local ethics committee of the University of Strasbourg (i.e., Comité d'Éthique de Recherche).

**Experimental protocol**. The experiment used the Unity software (Unity Technologies, v. 2019.3.9f1) to create the virtual environment. The HTC Vive Eye Pro (HTC Corp.) headset and controllers were used to immerse the participants in VR. Participants wore both the EEG and VR headsets while sitting on a chair. The use of VR has several advantages. Firstly, VR allows a better trade-off between fully controlled experimental settings and ecological experience (e.g., 3D visual percepts) in comparison with 2D screen setups. Secondly, it increases the engagement of the participant in the task at hand. Thirdly, the embedded eye-tracking system in the VR headset allows researchers to easily track and record the gaze in the 3D space. Eye-tracking was used to trigger visual stimuli. Participants were instructed to fixate the warning signal without moving their eyes, and it was only after a time interval free of saccades and eye-blinks that the warning signal was switched off. This procedure improves the data quality of EEG recordings. Participants were not aware of the presence of the eye-tracker nor of the contingency between saccades/eye-blinks and the offset of the warning signal.

Each participant was immersed in a virtual room, facing four 3D robots, each with a light whose color and onset were manipulated (Fig. 1a). The experiment was composed of two intertwined tasks: a variable foreperiod task and an asynchrony detection task. The asynchrony detection task consisted of discriminating whether the lights of two robots appeared synchronously or asynchronously (using delays of 11 ms, 33 ms, or 66 ms). Participants performed four blocks of 120 trials for each task. The participant switched tasks every block to reduce boredom. At the end of each experimental block, a break was proposed to the participant to remove the VR headset. Given the research questions investigated in this article, only data collected from the variable foreperiod task are presented.

**Variable foreperiod task and stimuli**. In case the button was pressed before the go-signal, a warning sound was delivered to the participant signaling the incorrect response. The procedure consisted of 4 blocks of 120 trials, with each block comprising 60 trials with short (S) FP and 60 trials with (L) long FP. The experiment started with 25 training trials. The procedure excluded the possibility of having three same foreperiods in a row (i.e., SSS or LLL). The light of the robots was presented at a distance of 4 meters from the participant. These lights were located at 8° and 24° of visual angle from the center of the scene. The presentation of the temporal (T) and neutral (N) cue conditions was alternated, taking the form of NTNT ($N = 24$) or TNTN ($N = 22$). At the beginning of each block, the two robots used in the condition were relocated to the center of the scene and the two other robots were relocated on the sides (randomly on the right and left). At the beginning of each trial, a colored (during temporal cue blocks) or uncolored (during neutral cue blocks) robot was randomly selected to include the warning signal and the target light. The matching of the robot's color with the FP was randomly assigned for each participant: blue for the short FP and turquoise for the long FP, or the reverse.

**Behavioral analyses**. On the one hand, pressing the button before the onset of the target (i.e., anticipated responses) reveal the impulsivity[58] in the two groups. On the other hand, pressing the button after the onset of the target was used to compute the two indexes (i.e., benefits from the passage of time and temporal attention cue).

**Eye-tracking acquisition and analyses**. The binocular gaze position was monitored using the eye-tracking system (Tobii Ltd.) embedded in the VR headset at a sampling rate of 90 Hz and an estimated spatial accuracy between 0.5° and 1.1°. The particularity of such a system is that (1) it tracks the gaze position independently of head movements, (2) it provides estimations of the gaze location in the 3D space rather than on-screen 2D space, and (3) the calibration-free data recording for saccades analysis renders the measure non-intrusive. Here we analyzed the likelihood of small fixational saccades during the foreperiods, as previously investigated[39,41,59]. Saccades of all sizes were included, but due to the task requirements to fixate the stimulus area, most saccades were small (1.4° of visual angle on average).

First, the onsets of blinks were identified with the HTC SRanipal SDK, detecting blinks via individual eye openness. Because blinks were particularly rare events given to non-blinking requirements to trigger the offset of the warning signal, trials containing at least one blink occurring during the time-window of interest (i.e., −200–600 ms in trials with short FP; −200–1200 ms in trials with long FP) were discarded (5.3% of total trials). Trials with anticipated responses were also discarded (1.85 % of the data). Further analysis was performed on 455 and 450 remaining trials on average in VGPs and NVGPs, respectively. Second, raw data (i.e., the 3D gaze position over time) of each trial were interpolated with a spline method to increase the temporal precision followed by the calculations of the derivations of the speed of vertical and horizontal movements.

Saccades were detected using a modification of a published algorithm[60] based on gaze's velocities. A threshold criterion for saccades detection was determined in the 2D velocity space based on the horizontal and the vertical velocities of gaze movement. This 2D space represented a plane surface located at the stimulus area (i.e., the warning signal and target). This threshold was represented by a 2D ellipse. For each trial, we set the threshold to be six times the SD of the gaze velocity[39,41,61] using a median-based estimate of the SD. Small fixational saccades were defined when six or more consecutive velocity samples (i.e., a minimum of 6 ms) were observed outside the ellipse.

Per standard procedure, we controlled for corrective saccades following overshoots that could have been confused with saccades. Thus, saccades were discarded when separated by <50 ms from the preceding one. We verified that the velocity and the magnitude of the saccades were correlated ($r = 0.78$), thus confirming a low false alarm rate of the saccade detection algorithm[62].

Visual inspection of the data revealed relatively low saccade rates across subjects. Hence, the saccade time series were smoothed using a moving average window of 100 ms, as in ref. [48]. Preliminary analyses of the results with a moving window of 50 ms (as in ref. [40,59,63]) did not affect the data interpretation, other than reducing the signal/noise ratio. The detection of peak inhibition and rebound were restricted to the 100-300 ms and 300-500 ms time intervals following the offset of the warning stimulus to avoid local minima and maxima, respectively.

**EEG acquisition and analysis**. EEG activity was continuously collected using a Biosemi ActiveTwo 10–20 system with 64 active channels at 1024 Hz sampling rates and the ActiView software. The electrode offset was kept below 20 mV. The offset values were the voltage difference between each electrode and the CMS-DRL reference channels. EEG analyses were performed with MNE-Python v.0.22.0[64].

The Autoreject algorithm[65] was used to detect and repair artifacts. The motive for using this algorithm was to maximize the signal/noise ratio by adapting automatically the artifact detection parameters for each participant. It implements topographic interpolations[66] to correct bad segments. One participant was removed from EEG analysis due to an excessive number of artifacts in the recording. The procedure rejected a mean average of 36 trials (SD = 7) and 35 trials (SD = 8) in VGPs and NVGPs, respectively. A surface Laplacian filter was applied (stiffness $m = 4$, $\lambda = 10^{-5}$) to the data resulting in reference-free current source densities (CSD) which increase the spatial resolution of the signal and reduce the signal deformation due to volume conduction[67].

For the CNV analysis, the data were filtered with a 0.1 Hz high pass filter and a 30 Hz low pass filter. Then, the segmentation of the trials included a time interval starting 1200 ms before the offset of the warning signal and ending 700 ms and 1300 ms after the offset of the warning signal for the short and long FP, respectively. We selected the electrodes presenting the peak of the CNV component (i.e., electrodes C1, C2, Cz, CP1, CP2, CPz, as in ref. [32,39]) recorded over centro-parietal sites. CNV activities were then z-score normalized, using the mean average of the 200 ms interval before the offset of the warning signal.

For the analysis of the oscillatory activity, time-frequency representations were computed for each trial using a wavelet approach[68]. A family of Morlet wavelets (Gaussian-windowed complex sine wave) was built to perform the convolution via fast Fourier transform over each channel. The family of wavelets was parametrized to extract frequencies from 4–40 Hz. The number of cycles of wavelets was linearly-spaced, from 3 cycles for the lowest frequency to 10 cycles for the highest frequency. This precaution was used to keep a well-balanced trade-off between time and frequency resolution at each frequency. A baseline correction was applied to transform the signal amplitude into dB change, and then into normalized z-scores using the mean average of the 1000 ms interval before the offset of the warning signal. Trials were then re-segmented to remove edge artifacts, starting 100 ms before the offset of the warning signal and finishing at 600 ms and 1200 ms after the offset of the warning signal for the trials with short and long FP, respectively. Preliminary visualization of the oscillatory activities across the scalp revealed two main temporo-spatial clusters, namely a power increase in the theta (4–8 Hz) and alpha (8–12 Hz) bands recorded in fronto-central electrodes and a power decrease in the alpha and beta (16–24 Hz) bands recorded over the left motor electrodes (Supplementary Fig. 4). Theta-band analysis was based on the medial fronto-central electrode FCz, where the presence of the oscillations was maximal (as in ref. [38]).

To quantify the phase-amplitude coupling, data-driven non-linear auto-regressive models[69] were used to build comodulograms. These comodulograms reflected the influence of the phase of theta-band oscillations recorded over the medial fronto-central cortex (electrode FCz) on the amplitude of the beta-band and gamma-band (24–40 Hz) oscillations recorded over the left motor cortex (electrode C3, similar to ref. [38]). For each participant and each cueing condition, comodulograms were computed from the current source densities of long FP trials using the entire 0–1000 ms time interval following the offset of the warning stimulus, providing a phase and amplitude frequency resolution of 0.2 Hz and 1 Hz, respectively.

**Statistics and reproducibility**. R (v.3.6.1) and the rstatix (v. 0.6.0) package were used to perform two-sided repeated-measures analyses of variances (rANOVA) and planned comparisons analysis with Tukey's HSD tests corrected for multiple comparisons with the false-discovery rate method[70]. For statistical analysis, EEG

data were downsampled to 512 Hz to facilitate computations. All rANOVAs were performed with a Greenhouse-Geisser correction when within-subject factors (Cue, Foreperiod, and Block) violated the sphericity assumption. Shapiro's test was used to evaluate the normal distribution of the data. Pearson's or Spearman's correlation analyses were used depending on the normality of the distribution of the data. To achieve the recommended 80% statistical power at $\alpha = 0.05$ and a small effect size $\eta^2_p = 0.1$ (as found for the benefit from the passage of time) for interaction effects, 18 participants per group would be required (computed using G*Power 3.1[71]). We increased this estimated sample to 23 participants to account for the low signal-noise ratio typical of EEG recordings. Additional Bayesian statistics have been conducted using Jamovi (v.1.6).

**Reporting summary**. Further information on research design is available in the Nature Research Reporting Summary linked to this article.

## Data availability

Source data[72] underlying figures are available in the open science framework (OSF), accessible at https://osf.io/54pj7/. Raw EEG and eye-tracking (source) data can be requested from the corresponding author.

## Code availability

Codes are available in the open science framework (OSF), accessible at https://osf.io/54pj7/. Python-MNE (v.0.22.0) was used to perform initial EEG data processing. R (v.3.6.1) were used to perform all data visualizations and statistical analyses.

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

## Acknowledgements

We would like to thank Profs. Hafed Ziad and Engbert Ralf for their valuable comments on the eye-tracking data. This study was funded by the EU, Horizon 2020 Framework Program, FET Proactive (VIRTUALTIMES consortium, grant agreement Id: 824128 to A.G.). Also, we would like to thank Dr. Clara Alida Cutello and Raquel Neal for their valuable comments.

## Author contributions

Conceptualization, methodology and writing—original draft, F.R.F. and A.G.; software, investigation and visualization F.R.F.; writing—review and editing, F.R.F., M.C., A.B., and A.G.; funding acquisition, resources and supervision, A.G. All authors approved the final version of the manuscript for submission.

## Competing interests

The authors declare no competing interests.
