## [Peer Review File · Communications Biology]

Reviewers' comments:

Reviewer #1 (Remarks to the Author):

Review of "Neurocognitive evidence of enhanced implicit temporal processing in video game players"

Manuscript Overview: The authors present the results of a cross-sectional/extreme groups study in which individuals who, as part of their daily lives, consistently play one form of video game (action video games) were contrasted against individuals who do not consistently play video games, on a speeded reaction time task while eye-movements and EEG were recorded. Specifically, in the task, participants viewed a robot presented in a VR environment. A light on the robot turned green to warn the participant the trial was beginning. Then either 400 ms or 1000 ms later, the light turned red. The participants' task was to press a button as quickly as possible when the light turned red. In some trials, the color of the robot indicated whether the light would turn red after 400 ms or 1000 ms (i.e., the timing was cued), while on other trials no cue as to when the red light would turn on was provided. The authors found that action game players benefitted more (RTs were disproportionately faster) in for the longer cued time (1000 ms) under the neutral cue as compared to the non-gamers. The authors found both oculomotor and EEG correlates of these behavioral shifts as well.

Review Overview: The manuscript has some substantial positives, including the mixed methods (behavior, eye-tracking, EEG) which is reasonably rare in this field (many papers contain two of the three, but few have all three), and associated results. For these reasons, I think the paper could make an impact on the field. However, there are some negatives that would preclude me from recommending the manuscript in its current form. Some of these are somewhat minor and can be addressed via shifts in narrative/discourse (e.g., some of the background literature review could be improved, a discussion of limitations is needed, etc.). However, some are more major related to analyses. For instance, the task the authors present is inherently a learning task, but analyses are not conducted/reported in such a way that takes that into account. There is also a question about whether the presence of an outlier is causing the violation of the assumptions of the tests that were used and whether the study is properly powered for the types of interactions that are being examined (which are often 3-way interactions).

Specific Comments:

Introduction:

1) Line 37-38. It's not clear to me why the authors start (2nd sentence of the manuscript) with internet gaming disorder? It seems like a really odd non-sequitur. It may be the authors were attempting to provide some type of "balance"? But it's not really clear that's needed since the literature on IGD is completely orthogonal to the literature on action games (and it's not even clear the same games are considered since the literature on IGD tends to mix all games together).

2) Line 39. I think it's probably a disservice to the field to say that the origins of action game effects remain "largely unknown." There is quite a lot of scholarship and theory on this point (e.g., related to changes in attentional control).

3) Line 41 – and going forward: I think the "implicit" versus "explicit" terminology is a bit problematic. Those uses tend to go with the extent to which knowledge is declarative (i.e., the knowledge of how to ride a bicycle is more implicit because it's difficult to explicitly describe what is known; the knowledge of the capital of Portugal is explicit because it can be explicitly declared). It seems like the authors throughout mean something more akin to "internal" versus "external" cues or something like that. I would definitely avoid the implicit/explicit language because I think it'll confuse readers.

4) Line 42 – I'm not sure what the authors mean by "the temporal sequences of stimuli in these games are repeated from one match to another." It's true that in some action games, sequences are

repeated (e.g., games with linear story modes). But it's frankly increasingly rare to see this in modern action games where there is more stochasticity put in enemy behavior (i.e., I think it's fair to say that interacting with fully predictable temporal sequences make up a tiny tiny minority of time action gamers spend playing). Furthermore, the use of "match" in this context makes it seem like the authors are referencing games that are competitive against other people – in which case it seems unlikely that any temporal sequences would ever be repeated.

This is actually a really important point because it gets at the heart of why the authors think that action games might be useful in this regime. If the authors think that repeated sequences are important, then it seems like the methods would need to reflect that (i.e., select out gamers that play games with repeated sequences). At a minimum it would be useful to give examples of the types of gameplay that match the experiences that they believe will cause a shift in performance on their task.

Similarly on line 47-48 when the authors say, "[action gamers] might explicitly use a visual cue to predict the exact moment of targets in their visual field" – this mechanism is indeed quite a bit more common in action games (e.g., color changes that indicate vulnerability of enemies or something like that), the exact cues would be completely embedded in a particular game. As such, there's no way for this knowledge to directly transfer to a new task like the ones the authors use. The best that could be done is for action gamers to learn how to detect the relationship between cues and timing. Which would necessitate a learning-based analysis to examine (see comments on results).

5) Line 49 – I don't think that this line: "except for one investigation, these studies explored spatial but not temporal aspects of attention" is accurate. For instance, there are multiple studies in the field that have examined performance in a host of more temporal tasks:

<https://www.nature.com/articles/nature01647>
<https://www.sciencedirect.com/science/article/abs/pii/S0028393209000657?via=ihub>
<https://link.springer.com/article/10.3758/s13414-011-0194-7>
<https://link.springer.com/article/10.3758/APP.72.4.1120>
<https://www.sciencedirect.com/science/article/abs/pii/S000169180500020X?via=ihub>
<https://link.springer.com/article/10.1007/s41465-017-0021-8>
<https://www.sciencedirect.com/science/article/pii/S004269890900474X>

So this literature definitely needs to be considered with respect to framing the hypotheses and results.

Line 50) This line: "In addition video games have been proposed as a potential rehabilitation tool for psychiatric disorders" also feels like a non-sequitur unless the authors can tie their research questions more strongly to psychiatric disorder (i.e., the work that is cited there doesn't link that well).

Results:

There are three large-ish issues with the results/analyses as presented:

1) As noted above, the task employed by the authors is inherently a learning task. There is no way for participants to know that just two start-cue  target-appearance values are used throughout or what those values might be. Furthermore, there is no way for them to know how the color of the robots might indicate the timings. Given this, **all** of the analyses should be modeling the impact of time – ideally at the participant level. In some analyses right now the authors do include "block" as a factor (though they don't report whether any effects go with block). But it's not clear that "block" is the right unit of analysis – and more to the point – I would argue it's almost certainly not. In essence, by using "block" in an analysis, the authors are indicating that they believe that the 120 trials in each block are generated iid (within condition of course). But I'd be shocked if there wasn't significant learning within blocks – in particular within the first block. After all, the statistical regularities in the task aren't likely to be super hard to learn (2 timings, 2 color  timing relations), in which case they could be learned quite quickly. But by combining all the trials in Block #1 together, the authors are smoothing over any of this.

My own preference would be for the authors to model the data in a time-continuous fashion (e.g., see <https://alab.psych.wisc.edu/papers/files/Cochraneetal2018.pdf>) at the individual level. But because that type of analysis is pretty complicated and not every group has the capacity for them, at a minimum, the authors would need to make a case for either why their units of analysis are appropriate given that learning is likely taking place (or that they show no learning takes place, which again, seems impossible given the task).

This is particularly important since one of the major theories in the field posits that action gamers learn statistics more quickly than non-gamers, which is partially why they show an advantage on tasks.

2) The second large-ish issue is that it's not clear to me that the study is properly powered for 3-way interactions. Thus, I'm not sure what we learn from three-way interactions that are not significant. At a minimum, the authors might want to consider some quantitative metrics of the "strength" of their non-significant findings (i.e., are these equivocal or strongly in favor of the null). They also need to be *very* careful about drawing inferences from patterns of significant versus non-significant analyses.

3) There is at least one *very* large outlier in the NVGP data set (my guess is that it's just one person – see figure S1). Including that individual in the analyses that the authors have conducted has the potential to mess up every estimated value. At a minimum it would be important to know if the results hold with that one individual (assuming it is one individual) removed.

Other minor things:

Line 87: The authors write, "VGPs are believed to be impulsive (but see 41)" to start their results section. (1) If the authors are going to write that VGPs are believed to be impulsive, it would either be useful to provide some type of citations or at least say where that belief is situated (i.e., "amongst the general public"?). (2) It's not clear that citation 41 is a good example of a "but see". The authors should look at: <https://journals.sagepub.com/doi/10.1111/j.1467-8721.2009.01660.x>

Which not only discuss this issue, but utilize a task designed to test impulsivity.

Line 87: The question of impulsivity is a major one, so I'm not sure why the authors chose to put those results in the supplement? Readers are definitely going to wonder if the gamers are just more impulsive and so saying that, if anything, there are fewer anticipatory errors, would be important to get out of the way right away in the manuscript proper.

Discussion:

1) One aspect that needs to be included in the discussion is a consideration of the limitations of correlational study designs in this field (i.e., with regard to an inability to draw strong causal inferences). The authors also need to be careful with causal language throughout for this same reason (e.g., avoiding terms like "practice with" that suggest that the data indicates the relationship has been shown to be causal).

2) The discussion should also speak to the issues I raised above with respect to key ingredients in action video games. For instance, if the authors believe that deterministic timing sequences that can be learned are important, they should state that and then also consider what other types of games have those in them (there are many).

Methods:

1) The authors would need to be more clear about how their samples were recruited (e.g., were the participants aware that their participation was predicated on them being an action gamer or a non-gamer)? See the discussion of overt versus covert recruiting in reference 7 in the current manuscript.

Reviewer #2 (Remarks to the Author):

Evaluation of the manuscript: Commun Biol 11992

This is an interesting cross-sectional study focused on characterizing how intensive practice with (action) video games might improve explicit and implicit temporal processing. The study has several merits, including the multifaceted methodological approach which included virtual reality, eye-tracking, behavioral data, electroencephalography (analyzed both in terms of ERPs and advanced time-frequency approaches). The results are interesting and the article is generally well written. I have some minor concerns especially regarding data interpretation.

Minor points

- Some very small effect sizes need to be commented (e.g., $\eta^2p = .0008$)
- It is bit annoying that some potentially interesting statistical results (e.g., those related to peak of saccadic rebound) are reported only in the Supplementary material, as it would take a similar amount of characters to report these values directly in the manuscript rather than substituting them with "see Supplementary Results".
- Page 6, line 245 "a benefit most likely resulting from action video game practice". I am not sure here about the causal interpretation offered by the authors about the differential implicit temporal preparation benefit in video-game players vs controls. What about the possibility of self-selection? That is, individuals with better implicit temporal preparation capacities self-select themselves as action video game players. To make such strong assertions ("most likely resulting from action video game practice") one needs to run an experimental longitudinal study. The suggested interpretation by the authors, although plausible, should be put forward with more caution and with the suggestion of such longitudinal investigations in the future. Even when later on (page 7, line 285 etc.) the authors interpret the group differences as due to transfer learning mechanisms, this is also an inference that over-interprets the current data and needs caution as well when it is stated, and not only later on when finally the authors prudentially wrote (lines 287-288): "We deem that further work should evaluate the causal effect of action video game play on implicit temporal processing". This caution note is great but also previous statements of causal inference should be toned down.
- Methods – Participants (page 7, from line 292). Please report more details regarding the recruited samples if available (e.g., cultural background, education level, recruitment procedure etc.).
- Page 7, line 299: given that a portion of control participants had no extensive practice ever in action video game ($N = 16$), it could be useful to also run a control re-analysis of the between group effects with this control sub-group only (to exclude even further potential long-lasting transfer effects of action video game playing in the control group).
- Lines 311-313: "it was only after a time interval free of saccades and eye-blinks that the warning signal was switched off. This procedure improves the data quality of EEG recordings". This procedure is of course useful but probably not without consequences. For instance, it could have made the timing of the events more under the control of participants' eye-movement behavior, as a sort of neurofeedback-like phenomenon. Please also try to take care of this peculiarity of the experimental paradigm in the analyses (e.g., 1. by using the average number of eye-movement artifact for each participants; 2. by using a regressor of no interest in the data analysis which takes into account the waiting time before each warning signal switching off). More generally, it would be also advisable to analyze data with linear mixed models and single trial data inserted in the model. If the authors decide to go for this type of analysis, it would be useful to also include trials (rank order) to control for effects of learning or fatigue, and preceding RT to control for trial-by-trial RT autocorrelation (for a similar approach, see <https://doi.org/10.1016/j.neuroimage.2021.117867>)
- Page 9, line 372: "One participant was removed from EEG analysis due to an excessive number of artifacts". Please specify here which group this excluded participant belonged to.
- It might be also advisable to analyze (control for) non-strategic sequential foreperiod effects (i.e.,

the influence of the previous trial foreperiod length on the RTs of the current trial), that are largely ignored right now, but that typically explain a lot of variance in this type of paradigms (e.g., <https://doi.org/10.1016/j.cognition.2013.10.006>; <https://doi.org/10.3389/fnhum.2018.00017>).

Reviewer #3 (Remarks to the Author):

Summary: The main aim of the current experiment was to understand the behavioural and neural markers of temporal attention mechanisms in action video game players and non-action-videogame players. A large number of previous studies have focused on spatial attention mechanisms in gamers compared to non-gamers as well as in training studies, however temporal attention mechanisms are underexplored. To this aim, $N = 23$ action video game players and $N = 23$ non-action-video game players were asked to detect target robots in a virtual environment which appeared either after a short or a long SOA after a cue. In order to prepare for the target response, different coloured robots acted as a cue; for instance a blue and turquoise robot indicated that the target robot would appear after a long or a short time interval respectively, and a grey robot acted as a neutral cue which did not predict any target robot. Behavioural results indicated that gamers were faster than the non-gamers in the neutral cue condition, especially when the target robot followed the cue after a long time interval. "Estimates of the passage of time" indicated that gamers showed a benefit especially in the neutral cue condition. Furthermore, gamers were faster when the cue predicted a short interval compared to a long time interval. Oculomotor responses indicated a shorter latency of the peak of inhibition oculomotor response in VGPs. Neural responses operationalized as the amplitude of the contingent negative variation (CNV) did not significantly differ between gamers and non-gamers. Across all participants, a negative correlation was observed which indicated that those participants who had a steeper CNV also benefitted more from the temporal cue in trials with long foreperiod (FP). Furthermore, reduced theta oscillations have been observed in action video-game players compared to non-action video game players "when the probability occurrence of the target was indeed low". Additionally, an increased phase-amplitude coupling was observed in action video game players, especially in the neutral condition. The authors concluded that gamers showed advantages in the "implicit" passage of time (given likelihood of the appearance of the target) which were also documented in neural plastic changes.

The authors documented temporal advantages in gamers compared to non-gamers by referring to a large range of behavioural and neural signals. I think this is a very interesting study which includes a rich data set. However, I have several comments which need to be addressed. 1. The authors mentioned briefly in the discussion, that training studies need to be run in order to draw any causal conclusion about the impact of video game training on temporal behavioural and neural functions. The current study can only indicate that these effects have been found in gamers and non-gamers however, causal conclusions cannot be drawn - this needs to be addressed throughout the paper, also in the introduction (see comment below).

2. A lot of results have been presented but I find not all of them very convincing.

General: Did the authors also analyse accuracy or dprime? This needs to be mentioned.

Line 101 and following: VGPs may have learned to optimize the task performance more rapidly than NVGPs. – Could the authors also present the performance across the different blocks? This would indicate whether the groups did not differ from the start but that VGPs develop faster performance across the different blocks.

Line 106: Novel estimates of the passage of time and temporal orienting effects: I recommend to describe and further justify this formula. For instance why is this formula only considering short FP in the numerator and not the sum of both RTs; the analogue for the percentage of speed change.

Lines 111: and following: It is not clear to me why the t-test analysis "helps to better evidence the benefits of the passage of time".

Lines 114: Cue, Block and Group analysis: There was no interaction with the factor Block, which would indicate the passage of time and whether there are any advantages in the video game group. If there would be any effect in "passage of time" this should be documented in the block factor. However,

there is no interaction or main effect of block. The rationale of this analysis needs to be further documented.

Figure 3A: there is also an increased suppression in VGPs, is this significant?

Figure 3C: separate correlations in each group should be documented-is the correlation driven by outliers?

Figure 4f: Is the correlation driven by outliers? Also, is the correlation significant separately in each group? This would be more convincing:

However, a Pearson correlation analysis revealed that participants who benefited from the temporal cue in trials with long FP had a more negative CNV slope in trials with the temporal cue rather than with the neutral cue ($r = -.323$, $p = .031$, see Figure 4F). This result supports the literature suggesting that the CNV slope reflects the explicit temporal orienting phenomenon.

The authors focused on the CNV over central electrodes. Could the authors also provide clusters of more posterior electrodes as well as frontal electrodes?

Figure 6A and b: oscillation plots should be shown separately for each group (NVGPs and VGPS)

Methods

Participants: The VGP and NVGP group are not gender matched, and probably also not age matched. This needs to be controlled.

Line 587: How many trials in each block?

Were practice trials included?

Methods: how many trials are included in each group? For saccadic reaction times as well as ERP responses?

Figure 1 B: I think it would be helpful if the timeline is mentioned with regards to the two possible scenarios: short and long time period for the warning signal, that is two separate figures. This would help to follow the segmentation procedure for the CNV. For instance, the predictive cues are not included in the time line, but this would help to understand the segmentation.

Also, the authors talk about foreperiod – could more conventional terms such as interstimulus interval or SOA be used here?

Line 167: This time interval starts at the time of the earliest possible occurrence of the target and ends when the target appears.

Does it mean when the next target appears?

Results:

Line 87: VGPs are believed to be impulsive -here another study shows evidences that VGPs are not impulsive

Dye, M. W., Green, C. S., & Bavelier, D. (2009). Increasing speed of processing with action video games. *Current directions in psychological science*, 18(6), 321-326.

Abstract: Conclusion “may help remediate timing alterations in psychiatric populations” needs further clarification as the experiment and also the population is unrelated to any psychiatric disorder

Introduction: In the introduction the hypothesis of “learning to learn” and enhanced “attentional control functions” should be introduced as underlying principles why gamers outperform non-gamers in many different tasks. For instance, the principle of “learning to learn” might allow gamers to learn the perceptual templates faster than in video-game players (see also Bejjanki et al., 2014; Zhang et al., 2021). Enhanced attentional control functions and it’s neural correlates have been discussed to be one underlying mechanism of enhanced perceptual and attentional control functions.

Bavelier, D., Achtman, R. L., Mani, M., & Föcker, J. (2012). Neural bases of selective attention in

action video game players. *Vision research*, 61, 132-143.

Bejjanki, V. R., Zhang, R., Li, R., Pouget, A., Green, C. S., Lu, Z. L., & Bavelier, D. (2014). Action video game play facilitates the development of better perceptual templates. *Proceedings of the National Academy of Sciences*, 111(47), 16961-16966.

Föcker, J., Mortazavi, M., Khoe, W., Hillyard, S. A., & Bavelier, D. (2019). Neural correlates of enhanced visual attentional control in action video game players: an event-related potential study. *Journal of Cognitive Neuroscience*, 31(3), 377-389.

Green, C. S., Pouget, A., & Bavelier, D. (2010). Improved probabilistic inference as a general learning mechanism with action video games. *Current biology*, 20(17), 1573-1579.

Zhang, R. Y., Chopin, A., Shibata, K., Lu, Z. L., Jaeggi, S. M., Buschkuhl, M., ... & Bavelier, D. (2021). Action video game play facilitates "learning to learn". *Communications biology*, 4(1), 1-10.

The introduction gives the impression that a training study has been run – but this is not the case. We need to know which type of mechanisms are enhanced through gaming – as this is a cross-sectional study, causal conclusions cannot be drawn and the study also does not provide a unique answer to this question. Training studies are needed in order to answer the causal relationship here.

Line 57: "whether reaction time performance" – what about other dependent variables such as accuracy or dprime?

Line 49: "However, except for one investigation, these studies explored spatial but not temporal aspects of attention". Please also add: Green, C. S., & Bavelier, D. (2003). Action video game modifies visual selective attention. *Nature*, 423(6939), 534-537. The attentional blink task as an example about "attention over time" has been used here as well.

Response to Reviewers

Article: COMMSBIO-22-0592

Reviewer #1 (Remarks to the Author):

Review of “Neurocognitive evidence of enhanced implicit temporal processing in video game players”

Manuscript Overview: The authors present the results of a cross-sectional/extreme groups study in which individuals who, as part of their daily lives, consistently play one form of video game (action video games) were contrasted against individuals who do not consistently play video games, on a speeded reaction time task while eye-movements and EEG were recorded. Specifically, in the task, participants viewed a robot presented in a VR environment. A light on the robot turned green to warn the participant the trial was beginning. Then either 400 ms or 1000 ms later, the light turned red. The participants' task was to press a button as quickly as possible when the light turned red. In some trials, the color of the robot indicated whether the light would turn red after 400 ms or 1000 ms (i.e., the timing was cued), while on other trials no cue as to when the red light would turn on was provided. The authors found that action game players benefitted more (RTs were disproportionately faster) in for the longer cued time (1000 ms) under the neutral cue as compared to the non-gamers. The authors found both oculomotor and EEG correlates of these behavioral shifts as well.

Review Overview: The manuscript has some substantial positives, including the mixed methods (behavior, eye-tracking, EEG) which is reasonably rare in this field (many papers contain two of the three, but few have all three), and associated results. For these reasons, I think the paper could make an impact on the field. However, there are some negatives that would preclude me from recommending the manuscript in its current form. Some of these are somewhat minor and can be addressed via shifts in narrative/discourse (e.g., some of the background literature review could be improved, a discussion of limitations is needed, etc.). However, some are more major related to analyses. For instance, the task the authors present is inherently a learning task, but analyses are not conducted/reported in such a way that takes that into account. There is also a question about whether the presence of an outlier is causing the violation of the assumptions of the tests that were used and whether the study is properly powered for the types of interactions that are being examined (which are often 3-way interactions).

Our answer: we would like to thank the reviewer for his/her positive assessment of our manuscript, and for his/her very useful comments, that helped us to better specify the theoretical background as well as the results of our study.

Specific Comments:

Introduction:

1) Line 37-38. It's not clear to me why the authors start (2nd sentence of the manuscript) with internet gaming disorder? It seems like a really odd non-sequitur. It may be the authors were attempting to provide some type of “balance”? But it's not really clear that's needed since the literature on IGD is completely orthogonal to the literature on action games (and it's not even clear the same games are considered since the literature on IGD tends to mix all games together).

Response:

Indeed, we were attempting to provide some balance to the reader. Yet, we agree with the point you've made and decided to remove this assertion.

We only leave: “Within a few decades research showed that playing video games can enhance cognition” (lines 37-38)

2) Line 39. I think it's probably a disservice to the field to say that the origins of action game effects remain “largely unknown.” There is quite a lot of scholarship and theory on this point (e.g., related to changes in attentional control).

Response:

Thank you for the comment, you are right. We decided to modify the assertion and include the 'learning to learn' and 'attentional control' theories.

"Within a few decades research showed that playing video games can enhance cognition. This enhancement involves increased ability to learn on the fly⁴⁻⁶ and improved attentional control^{7,8}. Game-induced cognitive enhancements depend on the gameplay, and their extent remains unclear." (lines 37-39).

3) Line 41 – and going forward: I think the "implicit" versus "explicit" terminology is a bit problematic. Those uses tend to go with the extent to which knowledge is declarative (i.e., the knowledge of how to ride a bicycle is more implicit because it's difficult to explicitly describe what is known; the knowledge of the capital of Portugal is explicit because it can be explicitly declared). It seems like the authors throughout mean something more akin to "internal" versus "external" cues or something like that. I would definitely avoid the implicit/explicit language because I think it'll confuse readers.

Response:

We agree that the "implicit" versus "explicit" terminology can be confusing to the reader. However, we find it difficult not to use these terms because they are established terms in the field of timing, and are associated with the kind of manipulations we used (Coull & Nobre, *Curr Opin Neurobiol*, 2008; Coull et al, *Neuropsychologia* 2013; Wiener et al 2010, *Neuropsychologia*).

We think that the "implicit" vs "explicit" terminology will be necessary for the readers to correctly link the central concepts of time perception and time processing with this study. We now give a definition of explicit and implicit timing at the beginning of the introduction, and try to erase any ambiguity throughout the paper. Explicit timing refers to any task in which participants receive explicit instruction to process temporal information. In contrast, during implicit timing tasks, participants are unaware of processing time. Timing intervenes in many tasks incidentally, and in the manuscript, we now use the term 'incidental' rather than automatic, because 'incidental' is more neutral regarding the type of associated mechanisms, and closer to the experimental manipulations.

"Explicit timing refers to any task in which participants receive explicit instruction to process temporal information. In contrast, during implicit timing tasks, participants are unaware of processing time. Timing intervenes in many tasks incidentally." (lines 41-43).

We considered the possibility of using externally vs. internally driven temporal prediction. Mento used this terminology (but not in his latest papers), which is understandable given how the different aspects were tested. As a matter of fact, the operationalization of explicit vs. implicit timing relies on the use of cues for explicit timing (the cue obliges the participant to think about time), whereas no cue is used for implicit timing, in which case time affects performance incidentally and the participant does not have to think about time. The cue manipulation may give the impression of a distinction between externally vs. internally driven timing. However, we decided against using 'internally' vs. 'externally' driven timing because this terminology may induce another type of confusion, this time with endogenous vs. exogenous cues. Temporal cues are not exogenous and do not attract attention automatically towards a moment in time, like a spatial cue can. On the contrary a temporal cue has to be processed cognitively and is an endogenous cue. Besides, time has still to be processed internally, even after the cue. For this reason, we think that 'externally vs. internally driven' would be as ambiguous as implicit and explicit timing. We do hope our definition and clarifications here and in the text help to suppress any ambiguity.

4) Line 42 – I'm not sure what the authors mean by "the temporal sequences of stimuli in these games are repeated from one match to another." It's true that in some action games, sequences are repeated (e.g., games with linear story modes). But it's frankly increasingly rare to see this in modern action games where there is more stochasticity put in enemy behavior (i.e., I think it's fair to say that interacting with fully predictable temporal sequences make up a tiny tiny minority of time action gamers spend playing). Furthermore, the use of "match" in this context makes it seem like the authors are referencing games that are competitive against other people – in which case it seems unlikely that any temporal sequences would ever be repeated.

This is actually a really important point because it gets at the heart of why the authors think that action games might be useful in this regime. If the authors think that repeated sequences are important, then it seems like the methods would need to reflect that (i.e., select out gamers that play games with repeated sequences). At a minimum it would be useful to give examples of the types of gameplay that

match the experiences that they believe will cause a shift in performance on their task.

Response:

Thank you for the comment, as it allows us to specify more precisely the link between the gaming experience and the current experimental manipulations. We totally agree that in modern games, since the early 2000s, the stochasticity in the enemy's behaviors (when the player is an AI) makes most of the timing of events unpredictable. Yet, some behavior remains relatively well predictable.

The initial idea behind our study was that in most action video games (eg. FPS), participants wait for the appearance of an enemy. Even if the time of occurrence of this enemy is stochastic, the probability is high and games can be expected to optimize their preparation when waiting for the enemy to appear in most FPS games. In the article, we proposed the following example to understand the relationship between gameplay and our task:

“Indeed, in most first-person shooter games subjects expect and wait for enemies to appear. Even though the time of occurrence of the enemy is stochastic, the probability of an enemy occurrence is high, and gamers can prepare to react as fast as possible when the enemy finally appears. Players likely benefit from practice to refine temporal expectations of forthcoming targets, even if they are unaware of such expectations” (lines 45-48).

Similarly on line 47-48 when the authors say, “[action gamers] might explicitly use a visual cue to predict the exact moment of targets in their visual field” – this mechanism is indeed quite a bit more common in action games (e.g., color changes that indicate vulnerability of enemies or something like that), the exact cues would be completely embedded in a particular game. As such, there's no way for this knowledge to directly transfer to a new task like the ones the authors use. The best that could be done is for action gamers to learn how to detect the relationship between cues and timing. Which would necessitate a learning-based analysis to examine (see comments on results).

Response:

The response to the comment is addressed below (with comments about the results).

As a reminder, all participants were aware of the relationship cue-foreperiod, the duration of the two possible foreperiods, and practiced training trials. This is made clearer in the introduction.

5) Line 49 – I don't think that this line: “except for one investigation, these studies explored spatial but not temporal aspects of attention” is accurate. For instance, there are multiple studies in the field that have examined performance in a host of more temporal tasks:

<https://www.nature.com/articles/nature01647>

<https://www.sciencedirect.com/science/article/abs/pii/S0028393209000657?via=ihub>

<https://link.springer.com/article/10.3758/s13414-011-0194-7>

<https://link.springer.com/article/10.3758/APP.72.4.1120>

<https://www.sciencedirect.com/science/article/abs/pii/S000169180500020X?via=ihub>

<https://link.springer.com/article/10.1007/s41465-017-0021-8>

<https://www.sciencedirect.com/science/article/pii/S004269890900474X>

So this literature definitely needs to be considered with respect to framing the hypotheses and results.

Response:

We realize that the term “temporal aspects of attention” was misleading, as it may have meant developmental or learning effects.

Indeed, in the papers you have mentioned, several temporal aspects of attention are evaluated in gamers and controls, especially how attention evolves in time with training, or how it evolves in time with age. Even though many studies in the field employed tasks with varying SOA or ISI (ie.

“foreperiod” in our experiment), this manipulation was not at the heart of their research questions. For example, studies using the attentional blink or spatial cueing effects assessed temporal aspects of attention deployment/recovery. However, the spatial attentional manipulations make it difficult to assess implicit temporal mechanisms per se.

In short, we agree with you that we had to be more specific and rephrased as follows:

“A large number of studies revealed that playing action video games improves explicit spatial attention mechanisms, their evolution over time and development, or explicit time discrimination abilities 8,16–23. However, it is still unclear whether VGPs learn to implicitly benefit from the passage of time to better anticipate future targets.” (lines 51-54).

Line 50) This line: “In addition video games have been proposed as a potential rehabilitation tool for psychiatric disorders” also feels like a non-sequitur unless the authors can tie their research questions more strongly to psychiatric disorder (i.e., the work that is cited there doesn’t link that well).

Response:

We agree that explanations were missing there. In our lab, the main line of research concerns implicit and explicit temporal skills in patients with psychiatric disorders. Multiple evidence tightened up symptoms found in schizophrenia with deficient implicit temporal processing of sensory information (see this review published in may; <https://www.nature.com/articles/s44159-022-00038-y>). The idea of video games for rehabilitation in psychiatry has been the starting point of this study. The link between this study and what has been published in the schizophrenia literature is the EEG and behavioral results found in patients using the Variable Foreperiod Task.

Therefore, we added a (short) sentence to better explain the idea behind the study.

“We need to know which types of mechanisms are enhanced in video game players, to better understand the impact of video game, and how they may help pathological groups. For example, implicit temporal mechanisms appear to be impaired in patients with schizophrenia (see 24 for a review), and video games have been proposed as a potential rehabilitation tool for psychiatric disorders such as schizophrenia 25. Knowing how video game play shapes brain mechanisms and behaviors will help to adapt these rehabilitation tools to pathologies. (lines 54-58).

Results:

There are three large-ish issues with the results/analyses as presented:

1) As noted above, the task employed by the authors is inherently a learning task. There is no way for participants to know that just two start-cue  target-appearance values are used throughout or what those values might be. Furthermore, there is no way for them to know how the color of the robots might indicate the timings. Given this, *all* of the analyses should be modeling the impact of time – ideally at the participant level. In some analyses right now the authors do include “block” as a factor (though they don’t report whether any effects go with block). But it’s not clear that “block” is the right unit of analysis – and more to the point – I would argue it’s almost certainly not. In essence, by using “block” in an analysis, the authors are indicating that they believe that the 120 trials in each block are generated iid (within condition of course). But I’d be shocked if there wasn’t significant learning within blocks – in particular within the first block. After all, the statistical regularities in the task aren’t likely to be super hard to learn (2 timings, 2 color  timing relations), in which case they could be learned quite quickly. But by combining all the trials in Block #1 together, the authors are smoothing over any of this.

My own preference would be for the authors to model the data in a time-continuous fashion (e.g., see <https://alab.psych.wisc.edu/papers/files/Cochraneetal2018.pdf>) at the individual level. But because that type of analysis is pretty complicated and not every group has the capacity for them, at a minimum, the authors would need to make a case for either why their units of analysis are appropriate given that learning is likely taking place (or that they show no learning takes place, which again, seems impossible given the task).

This is particularly important since one of the major theories in the field posits that action gamers learn statistics more quickly than non-gamers, which is partially why they show an advantage on tasks.

Response:

As a reminder, participants were aware of the cue-foreperiod relationship and the duration of the foreperiods (as indicated line 75), but also practiced 25 training trials (now added in line 386). During the initial analysis of the behavioral data, we looked at the data in a continuous fashion (in fitting local polynomial regression curves, ie. loeess functions) to evaluate possible differences in learning and usage of the temporal cue between VGPs and NVGPs. Nothing came out of it, and we tried another approach.

We now added in the Supplemental Results a control analysis evaluating the reaction times across the Trial Blocks and Group.

“A control analysis was conducted to evaluate the possibility that reaction times performance was optimized faster over the course of the experiment in VGPs compared with NVGPs. To do so, a two-way repeated-measures ANOVA with the factors Group and Block was performed on the reaction times data. The analysis revealed a main effect of the Block ($F(3, 132) = 10.718, p < .0001, \eta^2p = .$

196). However, post-hoc analyses did not reveal significant differences between trial blocks (all $p > .095$). Crucially, no main effect of the Group ($p = .16$) nor interaction effect ($p = .065$) was reported (see Figure S1B). Hence, the improvement of the performance over the course of the experiment appears non-specific to the group of participants.” (lines 17-23).

A critical issue was also whether the benefit from the passage of time evolved across blocks, especially early in the task. To evaluate this issue, we artificially separated all trials in 10 sub-blocks and computed the benefit from the passage of time (see Figure S1C) for each participant and each sub-block. Again, while a main effect of the Trial Block was revealed, no main effect of the Group nor interaction effect was revealed (see lines 24-38 in the Supplemental Results).

We additionally used a Bayesian analysis to evaluate the reliability of the negative effects (see lines 78-91 in the Supplemental Results).

In sum, we found no evidence that VGPs' reaction times differed from the start of the experiment or that VGPs develop faster performance across the different blocks.

We now mention the lack of main and interaction effect with the factor Trial Block in the main text (lines 133-134), and refer to the Fig. S1C in supplemental material.

2) The second large-ish issue is that it's not clear to me that the study is properly powered for 3-way interactions. Thus, I'm not sure what we learn from three-way interactions that are not significant. At a minimum, the authors might want to consider some quantitative metrics of the “strength” of their non-significant findings (i.e., are these equivocal or strongly in favor of the null). They also need to be *very* careful about drawing inferences from patterns of significant versus non-significant analyses.

Response:

Thank you for your important comment.

To increase the statistical power, researchers can 1) multiply the number of observations per condition and 2) increase the number of participants (Brysbaert et al. 2019, Cognition). We favoured the first solution. In EEG studies, the amount of behavioral data is most often sufficient to perform two- or three-way ANOVAs, given the important number of observations used to perform statistical analysis of EEG data. Here, we recorded 120 observations for each factor (i.e. 2 delays) of each experimental variable (i.e. 2, Foreperiod and Cue), for a total of 480 observations for each participant.

In the manuscript, we perform three-way rANOVAs in order to include the variable Trial Block (a variable of interest in the field – but not in the scope of the present study). Yet, we found no effect of the Trial Block, neither on the benefit from the passage of time nor the benefit from the temporal cue, but we found interaction effects between the two other variables (Cue*Group and Foreperiod*Group).

To evaluate the robustness of these interaction effects (which are at the core of the paper), we re-evaluated the benefit from the passage of time and the benefit from the temporal cue using rANOVAs without the (non-significant) Trial Block variable.

When re-evaluating the benefit from the passage of time, we observed a main effect of the Cue ($F(1, 44) = 19.7, p < .0001, n^2_p = .310$) and an interaction effect between the Group and the Cue ($F(1, 44) = 4.7, p = .036, n^2_p = .097$).

When re-evaluating the benefit from the temporal cue, we observed a main effect of the Foreperiod ($F(1, 44) = 20.1, p < .0001, n^2_p = .315$) and an interaction effect between the Group and the Foreperiod ($F(1, 44) = 5.8, p = .021, n^2_p = .116$).

Thus, this re-analysis (not reported in the Supplemental Results) of the behavioral data lead to the same conclusions as reported in the manuscript.

To test the reliability of the non-significant findings in behavioral data analysis reported in the manuscript, we performed an additional Bayesian repeated-measure analysis using Jamovi, a free software implemented in R. Here, we used a model comparison approach. Within the Bayesian framework, the strength of the evidence supporting H0 or H1 is reflected in the value of the Bayes Factor (BF_{10}). Please find the interpretation of the BF values in the table below.

Interpretation of the Bayes Factor (figure from Van Doorn et al. 2019; Psychonomic Bulletin & Review)

First, we evaluated the evidence for or against the absence of effect of the Trial Block variable on the benefit from the passage of time. The analysis reports weak to moderate evidence for the absence of a main effect of the Trial Block ($BF_{10} = 0.223$), for the absence of interaction effects between the Trial Block and the Group ($BF_{10} = 0.369$), between Trial Block and the Cue ($BF_{10} = 0.210$), and for the absence of triple interaction effect between the Trial Block, the Group and the Cue ($BF_{10} = 0.349$).

Second, we evaluated the evidence for or against the absence of effect of the Trial Block variable on the benefit from the temporal cue. The analysis reports weak evidence for the absence of a main effect of the Trial Block ($BF_{10} = 0.557$), for the absence of interaction effects between the Trial Block and the Group ($BF_{10} = 0.283$), between Trial Block and the Foreperiod ($BF_{10} = 0.243$), and for the absence of triple interaction effect between the Trial Block, the Group and the Foreperiod ($BF_{10} = 0.294$). Overall, all analyses suggest an absence of the effect of the trial block on behavioral data.

This additional Bayesian analysis is now reported in the Supplemental Results.

3) There is at least one *very* large outlier in the NVGP data set (my guess is that it's just one person – see figure S1). Including that individual in the analyses that the authors have conducted has the potential to mess up every estimated value. At a minimum it would be important to know if the results hold with that one individual (assuming it is one individual) removed.

Response:

The reviewer is right, the outlier is a single participant.

However, the calculation of our indexes by-passes this problem.

This participant (subject 41) appears particularly slow. Yet, our indexes reflect the slopes between different conditions (FP1 vs FP2; Neutral vs Temporal Cue) for each participant. Consequently, the calculation of our indexes is also a way to normalize the data (subject 41 is not an outlier anymore). Yet, we checked the effects of our variables on the reaction times without this subject 41.

No difference was found in the ANOVA nor the planned comparisons investigating the interaction effects. Altogether, including or removing this participant from the analysis does not affect the effect of the group on the benefit from the passage of time and temporal orienting. This information is now added in the legend of the Figure S1.

Other minor things:

Line 87: The authors write, “VGPs are believed to be impulsive (but see 41)” to start their results section. (1) If the authors are going to write that VGPs are believed to be impulsive, it would either be useful to provide some type of citations or at least say where that belief is situated (i.e., “amongst the general public?”). (2) It's not clear that citation 41 is a good example of a “but see”. The authors should look at: <https://journals.sagepub.com/doi/10.1111/j.1467-8721.2009.01660.x>

Which not only discuss this issue, but utilize a task designed to test impulsivity.

Response:

We agree with the reviewer, we meant that the belief exists in the general public. Thank you for the reference. We added the reference and corrected the sentence.

“VGPs are believed to be impulsive (but see 43) amongst the general public.” (line 94)

Line 87: The question of impulsivity is a major one, so I'm not sure why the authors chose to put those results in the supplement? Readers are definitely going to wonder if the gamers are just more impulsive and so saying that, if anything, there are fewer anticipatory errors, would be important to get out of the way right away in the manuscript proper.

Response:

The question of impulsivity is important. Also, we agree that readers will wonder about impulsivity in VGPs vs NVGPs. Yet, impulsivity is not central to the research question of the article, because it cannot explain that VGPs tend to better prepare for a late target than NVGPs, but are not faster for early targets.

Thus, we now mention the lack of evidence of increased impulsivity in VGPs and refer to the supplemental results in the main text.

“A preliminary analysis of premature responses (anticipation errors – responding before the target appearance) revealed no evidence of increased impulsivity in VGPs compared with NVGPs (see Supplemental Results).” (lines 94-96)

Discussion:

1) One aspect that needs to be included in the discussion is a consideration of the limitations of correlational study designs in this field (i.e., with regard to an inability to draw strong causal inferences). The authors also need to be careful with causal language throughout for this same reason (e.g., avoiding terms like “practice with” that suggest that the data indicates the relationship has been shown to be causal).

Response:

Thank you for your comment. Throughout the paper, we removed assertions in sentences that could have been interpreted as causal inferences.

Also, we included the following statement in the conclusion:

“Unfortunately, the present study cannot draw strong causal inferences on the effect of game play on temporal cognition. We deem that further work should evaluate the causal effect of action video game play on the implicit processing of time using a longitudinal approach, ...” (lines 317-320).

2) The discussion should also speak to the issues I raised above with respect to key ingredients in action video games. For instance, if the authors believe that deterministic timing sequences that can be learned are important, they should state that and then also consider what other types of games have those in them (there are many).

Response:

As discussed above, we believe that learning to wait and prepare for targets through FPS games can explain the empirical evidence revealed in this study. Consequently, we stated in the conclusion this link between the learning of temporal sequences in these games and the evidence of increased implicit processing of time in VGPs.

“In most first-person shooter games, such as Call of duty, Counter Strike or Unreal Tournament series, temporal sequences require the player to expect successive events, track the time elapsing between these events, and prepare to allocate their attention in space and time accordingly. Here, we propose that being trained to expect and attend to successive events accurately in time helps to learn to benefit from the passage of time in general.” (lines 313-316)

Methods:

1) The authors would need to be more clear about how their samples were recruited (e.g., were the participants aware that their participation was predicated on them being an action gamer or a non-gamer)? See the discussion of overt versus covert recruiting in reference 7 in the current manuscript.

Response:

You are right, we forgot to include this important information. Participants were aware they were recruited depending on their playing activities. However, they were not aware of the goal of the study or the hypotheses, including the expected results. We now added this information in the Method section.

“In this cross-sectional study, participants were chosen using overt recruitment and screening criteria but were not aware of the specific aims of the study.” (lines 323-324).

Reviewer #2 (Remarks to the Author):

Evaluation of the manuscript: Commun Biol 11992

This is an interesting cross-sectional study focused on characterizing how intensive practice with (action) video games might improve explicit and implicit temporal processing. The study has several merits, including the multifaceted methodological approach which included virtual reality, eye-tracking, behavioral data, electroencephalography (analyzed both in terms of ERPs and advanced time-frequency approaches). The results are interesting and the article is generally well written. I have some minor concerns especially regarding data interpretation.

Response:

We would like to thank the reviewer for his/her positive assessment of our study and for his/her very useful suggestions that helped us to improve the paper.

Minor points

- Some very small effect sizes need to be commented (e.g., $\eta^2_p = .0008$)

Response:

Thank you very much for your comment. A mistake has been done in the previous report concerning the η^2_p . The R program was reporting generalized instead of partial eta-squared. We apologize for missing this error. All partial eta-squared have been updated and appear much higher than the generalized eta-squared.

- It is bit annoying that some potentially interesting statistical results (e.g., those related to peak of saccadic rebound) are reported only in the Supplementary material, as it would take a similar amount of characters to report these values directly in the manuscript rather than substituting them with "see Supplementary Results".

Response:

Thank you for your comment. You are right. Therefore, we inserted the text from the Supplementary material into the Manuscript.

"Similarly, the Group factor influenced the latencies of the peak rebound ($F(1, 44) = 4.18, p = .047; \eta^2_p = .087$): the peak occurred earlier in VGPs (Mean = 374 ms, CI = 11 ms) than NVGPs (Mean = 390 ms, CI = 12 ms). An interaction effect between the Group and the Cue ($F(1, 44) = 5.18, p = .043; \eta^2_p = .105$) indicated that the earlier peak rebound in VGPs relative to NVGPs was specific to the temporal cue condition ($p = .001$)."

- Page 6, line 245 "a benefit most likely resulting from action video game practice". I am not sure here about the causal interpretation offered by the authors about the differential implicit temporal preparation benefit in video-game players vs controls. What about the possibility of self-selection? That is, individuals with better implicit temporal preparation capacities self-select themselves as action video game players. To make such strong assertions ("most likely resulting from action video game practice") one needs to run an experimental longitudinal study. The suggested interpretation by the authors, although plausible, should be put forward with more caution and with the suggestion of such longitudinal investigations in the future. Even when later on (page 7, line 285 etc.) the authors interpret the group differences as due to transfer learning mechanisms, this is also an inference that over-interprets the current data and needs caution as well when it is stated, and not only later on when finally the authors prudentially wrote (lines 287-288): "We deem that further work should evaluate the causal effect of action video game play on implicit temporal processing". This caution note is great but also previous statements of causal inference should be toned down.

Response:

You are right, thank you. We specified throughout the paper that our group effects do not reflect causal relationships whatsoever. We add that longitudinal investigations are needed to evaluate the causal effect of video game on implicit temporal skills.

"Unfortunately, the present study cannot draw strong causal inferences on the effect of game play on temporal cognition. We deem that further work should evaluate the causal effect of action video game play on the implicit processing of time using a longitudinal approach, ..." (lines 316-318).

- Methods – Participants (page 7, from line 292). Please report more details regarding the recruited samples if available (e.g., cultural background, education level, recruitment procedure etc.).

Response:

More details about the recruitment procedure, the education level and estimates of the visual acuity of the samples, and t- and Chi2-tests comparing the age, gender, education level and visual acuity scores between VGPs and NVGPs have been added.

“In this cross-sectional study, participants were chosen using overt recruitment and screening criteria but were not aware of the specific aims of the study” (lines 323-324).

“T-test revealed no difference in age ($p = .312$) and Chi2 -test revealed no difference in gender ($p = .391$) between VGPs and NVGPs. However, the education level was slightly lower in VGPs (mean = 14.3 years) than NVGPs (mean = 15.7 years, $p = .012$). All subjects had normal or corrected-to-normal visual acuity, as checked with the Freiburg Visual Acuity Test 57 (estimates of visual acuity did not vary between groups, $p = .329$).” (lines 331-334)

The education level did not seem related to the benefits from the passage of time nor to the benefits from the temporal cue, as we now mention in the Results section:

“A control analysis revealed no correlation between the education level and the benefits from the passage of time, nor between the education level and the benefit from the temporal cue.” (lines 144-145)

- Page 7, line 299: given that a portion of control participants had no extensive practice ever in action video game ($N = 16$), it could be useful to also run a control re-analysis of the between group effects with this control sub-group only (to exclude even further potential long-lasting transfer effects of action video game playing in the control group).

Response:

Thank you for this suggestion.

We performed a control re-analysis on the indexes representing the benefit from the passage of time (where the most important effect was found in the study) and the temporal cue with the NVGPs with no extensive practice ever (let's call them the NVGP_{at_all} group)

We performed a two-way ANOVA with the factors Group (VGPs vs NVGP_{at_all} ($n=16$)) and Cue on the benefit from the passage of time. Only the main effect of the Cue was revealed ($p < .0001$).

Similarly, we performed a two-way ANOVA with the factors Group (VGPs vs NVGP_{at_all} ($n=16$)) and Foreperiod on the benefit from the temporal cue. Here, only the main effect of the Foreperiod was revealed ($p < .0001$). Importantly, the lack of significant interaction effects could relate to the smaller sample size inducing weaker statistical power.

Hence, it is difficult to infer whether long-lasting transfer effects of video game practice affected the benefit from the passage of time or the benefit from the temporal cue in NVGPs.

Given the small sample size used for this analysis, we preferred to not add this analysis to the manuscript.

- Lines 311-313: “it was only after a time interval free of saccades and eye-blinks that the warning signal was switched off. This procedure improves the data quality of EEG recordings”. This procedure is of course useful but probably not without consequences. For instance, it could have made the timing of the events more under the control of participants' eye-movement behavior, as a sort of neurofeedback-like phenomenon. Please also try to take care of this peculiarity of the experimental paradigm in the analyses (e.g., 1. by using the average number of eye-movement artifact for each participant; 2. by using a regressor of no interest in the data analysis which takes into account the waiting time before each warning signal switching off). More generally, it would be also advisable to analyze data with linear mixed models and single trial data inserted in the model. If the authors decide to go for this type of analysis, it would be useful to also include trials (rank order) to control for effects of learning or fatigue, and preceding RT to control for trial-by-trial RT autocorrelation (for a similar approach, see <https://doi.org/10.1016/j.neuroimage.2021.117867>)

Response:

We agree that the required time interval free of saccades could have had consequences. However, in our experiment participants were not aware of the contingency between saccades/eye-blinks and the cue offset (they were not even aware of the presence of the eye-tracking system). We added this comment in the Manuscript (lines 347-348).

Yet, we had indeed conducted an analysis (that was not included in the article) to verify the effect of the duration of the 'time interval free of saccades and eye-blinks' on the behavioral results (ie. reaction times and benefit from the passage of time and the temporal cue). No effect was revealed.

Hence, it does not seem to have induced substantial consequences on participants' behaviors (see Figure below).

Effect of the duration of the pre-cue interval on reaction times.

We know the great advantages of using linear mixed models (LMM) for data analysis. Unfortunately, multiple experiences with LMM pushed us to use a more traditional ANOVA approach. In previous research, we always faced difficulties to make LMM converge during data fitting (hence preventing reliable statistical results and reproducibility of the experimental results. This problem seems frequent with complex models, and how to appropriately parametrize these models does not seem clear to us given the literature (see the debate in Barr et al. 2013, Journal of Memory and Language; Bates et al. 2015, arXiv but also Eager and Roy, 2017, arXiv).

Given that both reviewers 1 and 2 asked for learning/fatigue effects, we added a graph illustrating the evolution of the hazard function over time in supplementary material (see Figure S1) and details about its analysis.

- Page 9, line 372: "One participant was removed from EEG analysis due to an excessive number of artifacts". Please specify here which group this excluded participant belonged to.

Response:

Thank you for your remark. We now specified the group (VGP) in the manuscript.

"One VGP has been removed from the EEG analysis due to excessive noise in the recorded signal." (line 334-335).

- It might be also advisable to analyze (control for) non-strategic sequential foreperiod effects (i.e., the influence of the previous trial foreperiod length on the RTs of the current trial), that are largely ignored right now, but that typically explain a lot of variance in this type of paradigms (e.g., <https://doi.org/10.1016/j.cognition.2013.10.006>; <https://doi.org/10.3389/fnhum.2018.00017>).

Response:

Great remark, thank you. We were aware of these effects. We added a paragraph and a Supplemental Figure (Fig. S2) in the Supplemental Results (lines 57-72).

Reviewer #3 (Remarks to the Author):

Summary: The main aim of the current experiment was to understand the behavioural and neural markers of temporal attention mechanisms in action video game players and non-action-videogame players. A large number of previous studies have focused on spatial attention mechanisms in gamers compared to non-gamers as well as in training studies, however temporal attention mechanisms are underexplored. To this aim, N = 23 action video game players and N = 23 non-action-video game players were asked to detect target robots in a virtual environment which appeared either after a short or a long SOA after a cue. In order to prepare for the target response, different coloured robots acted as a cue; for instance a blue and turquoise robot indicated that the target robot would appear after a long or a short time interval respectively, and a grey robot acted as a neutral cue which did not predict any target robot. Behavioural results indicated that gamers were faster than the non-gamers in the neutral cue condition, especially when the target robot followed the cue after a long time interval. "Estimates of the passage of time" indicated that gamers showed a benefit especially in the neutral cue condition. Furthermore, gamers were faster when the cue predicted a short interval compared to a long time interval. Oculomotor responses indicated a shorter latency of the peak of inhibition oculomotor response in VGPs. Neural responses operationalized as the amplitude of the contingent negative variation (CNV) did not significantly differ between gamers and non-gamers. Across all participants, a negative correlation was observed which indicated that those participants who had a steeper CNV also benefitted more from the temporal cue in trials with long foreperiod (FP). Furthermore, reduced theta oscillations have been observed in action video-game players compared to non-action video game players "when the probability occurrence of the target was indeed low". Additionally, an increased phase-amplitude coupling was observed in action video game players, especially in the neutral condition. The authors concluded that gamers showed advantages in the "implicit" passage of time (given likelihood of the appearance of the target) which were also documented in neural plastic changes.

The authors documented temporal advantages in gamers compared to non-gamers by referring to a large range of behavioural and neural signals. I think this is a very interesting study which includes a rich data set. However, I have several comments which need to be addressed.

Response:

We would like to thank the reviewer for his/her positive assessment of our study and for raising important points that helped us to improve the paper.

1. The authors mentioned briefly in the discussion, that training studies need to be run in order to draw any causal conclusion about the impact of video game training on temporal behavioural and neural functions. The current study can only indicate that these effects have been found in gamers and non-gamers however, causal conclusions cannot be drawn - this needs to be addressed throughout the paper, also in the introduction (see comment below).

Response:

You are right, thank you. We corrected the paper to make sure we never suggest that our group effects reflect causal relationships. We emphasize this point more strongly in the discussion. "Unfortunately, the present study cannot draw strong causal inferences on the effect of game play on temporal cognition. We deem that further work should evaluate the causal effect of action video game play on the implicit processing of time using a longitudinal approach, ..." (lines 316-318).

2. A lot of results have been presented but I find not all of them very convincing.

General: Did the authors also analyse accuracy or dprime? This needs to be mentioned.

Response:

Indeed, accuracy has been analyzed as anticipation errors (lines 94-96 in Manuscript; lines 3-13 in Supplemental Results). We reported the main effects of the Group (less errors in VGPs than NVGPs), the Cue (less errors in temporal than neutral cueing conditions) and the Foreperiod (less errors in short than long FP).

However, d' values could not be calculated in such a task. Our task is very easy and induce a very low percentage of errors.

Line 101 and following: VGPs may have learned to optimize the task performance more rapidly than NVGPs. – Could the authors also present the performance across the different blocks? This would

indicate whether the groups did not differ from the start but that VGPs develop faster performance across the different blocks.

Response:

Thank you for the comment. We agree this is an important point and we specify the results of our investigations in the supplementary material.

Our analyses of the indexes of benefits from the passage of time and temporal cueing included a Trial Block (1, 2, 3 or 4) factor. No effect of the Trial Block was revealed in these analyses. We now add a figure in the Supplemental Results (Figure S1C) illustrating the evolution of the benefit of the passage of time across blocks.

We also added in the Supplemental Results a control analysis evaluating reaction times across the Trial Blocks and Group (Figure S1B). While a main effect of the Trial Block was revealed, no main effect of the Group nor interaction effect was revealed.

“A control analysis was conducted to evaluate the possibility that reaction times performance was optimized faster over the course of the experiment in VGPs compared with NVGPs. To do so, a two-way repeated-measures ANOVA with the factors Group and Block was performed on the reaction times data. The analysis revealed a main effect of the Block ($F(3, 132) = 10.718, p < .0001, \eta^2_p = .196$). However, post-hoc analyses did not reveal significant differences between trial blocks (all $p > .095$). Crucially, no main effect of the Group ($p = .16$) nor interaction effect ($p = .065$) was reported (see Figure S1B). Hence, the improvement of the performance over the course of the experiment appears non-specific to the group of participants.” (lines 17-23 in Supplemental Results).

We also proposed an additional analysis of sub-blocks to evaluate the benefit from the passage of time over the course of the experiment (see lines 24-38 in Supplemental Results). No main nor interaction effect of the Group was reported.

Finally we added a Bayesian analysis to evaluate the reliability of the negative effects (also in the Supplemental Results, lines 78-91).

Consequently, in reply to your important remark, we found no evidence that VGPs' reaction times differed from the start of the experiment or that VGPs develop faster performance across the different blocks. Also, we preferred to perform a supplementary analysis rather than adding the factor Trial Blocks to the main analysis because the statistical power would have become too weak.

Line 106: Novel estimates of the passage of time and temporal orienting effects: I recommend to describe and further justify this formula. For instance why is this formula only considering short FP in the numerator and not the sum of both RTs; the analogue for the percentage of speed change.

Response:

We considered the short FP in the numerator because we wanted to evaluate how faster a participant can be relative to a specific condition (short FP) rather than the average speed. However, we agree this choice is debatable. We did both analyses, found similar results, and now mention this point in the manuscript.

“The $RT_{\text{short FP}}$ was entered in the denominator to evaluate how faster a participant can be in a specific condition (long FP) relative to a baseline condition (short FP) rather than the average response speed.” (lines 120-121)

“A control analysis revealed similar observations when considering the sum of the two conditions as denominators.” (lines 121-122)

Lines 111: and following: It is not clear to me why the t-test analysis “helps to better evidence the benefits of the passage of time”.

Response:

Thank you for allowing us to clarify this statement, which was misleading. We wished to emphasize that the calculation of speed change better captures the benefit from the passage of time than the more typical comparison of RT at short and long delays. We reformulated our sentence, and hope it is now clearer:

“A one-sample t-tests analysis conducted on the percentage of speed change revealed that all participants, independently of their group, benefited from the passage of time in both the neutral (all $p < .0001$) and temporal (all $p < .0015$) cue conditions. Given such benefit was not observed in the typical comparisons of $RT_{\text{short FP}}$ vs. $RT_{\text{long FP}}$ these results suggest our calculation is best suited to evidence the benefits from the passage of time.” (lines 122-126).

Lines 114: Cue, Block and Group analysis: There was no interaction with the factor Block, which would indicate the passage of time and whether there are any advantages in the video game group. If there would be any effect in “passage of time” this should be documented in the block factor. However, there is no interaction or main effect of block. The rationale of this analysis needs to be further documented.

Response:

The benefit of the passage of time is evaluated at the level of each trial by the RT advantage for long relative to short delays. Since VGPs show a larger RT advantage, we examined whether this was because they learned to benefit from the passage of time, more so than NVGPs.

The rationale of this analysis relates to the extensive work of Bavelier’s lab and colleagues (eg. Dye, M. W.G.) suggesting an enhanced learning ability in video game players at extracting and mastering key components of the tasks at hand. We consider this hypothesis in evaluating how the performance might have varied across the duration of the experiment (Trial Blocks 1-2-3-4).

This point is now clarified throughout the corrected manuscript.

In the present task, we found no evidence that the benefit from the passage of time (at the level of trials) varied across the experiment (as discussed in the **Response** below). We added a figure illustrating this point in supplementary material (Figure S1C). This could be explained by the fact that participants knew about the existence of only two foreperiods, by the use of the training phase and/or by the simplicity of the task at hand. It may also be possible that VGPs are generally better at benefitting from the passage of time, and that this advantage pre-existed to the participation to the study.

The rationale is further discussed in the Manuscript.

“We then performed a three-way rANOVA with the factors Cue, Block and Group to assess whether VGPs took better advantage of the passage of time than NVGPs across the cueing conditions and the duration of the experiment” (lines 126-128).

“No main effect ($p = .334$) nor interaction effects with the factor Block (all $p > .332$) was reported, as further illustrated in Supplementary Figure 1C” (lines 133-134).

Figure 3A: there is also an increased suppression in VGPs, is this significant?

Response:

We are not sure what you are referring to as “as increased suppression”. Our analysis shows shorter latencies of saccade rates at both peak inhibition (suppression?) and peak rebound.

Yet, we performed an additional analysis on the amplitude of the saccade rates at the peak inhibition and peak rebound. No significant effect was revealed. We now added this information to the manuscript.

“There was no significant group effect on the amplitude of saccade rates at peaks.” (lines 173-174).

Figure 3C: separate correlations in each group should be documented-is the correlation driven by outliers?

Response:

Thank you for your important remark. We verified that the correlation was indeed not driven by outliers. The correlation was specific to the VGP group. We corrected the text in the manuscript and the Figure 3C.

“Interestingly, a Pearson correlation revealed a significant negative correlation between the latency of the peak inhibition and the benefit from the passage of time ($r = -.46$, $p = .028$) in VGPs but not in NVGPs ($r = -.04$, $p = .861$), independently of the cueing condition.” (lines 168-170)

Figure 4f: Is the correlation driven by outliers? Also, is the correlation significant separately in each group? This would be more convincing:

However, a Pearson correlation analysis revealed that participants who benefited from the temporal cue in trials with long FP had a more negative CNV slope in trials with the temporal cue rather than

with the neutral cue ($r = -.323, p = .031$, see Figure 4F). This result supports the literature suggesting that the CNV slope reflects the explicit temporal orienting phenomenon.

Response:

We verified the correlation and it was not driven by outliers. The analysis did not provide a reliable correlation in each group, only the correlation combining all data (VGPs & NVGPs) was significant (possible due to the narrow distribution of VGPs' data points). Therefore, we provided correlations for both groups and combined data (Figure 4F).

The authors focused on the CNV over central electrodes. Could the authors also provide clusters of more posterior electrodes as well as frontal electrodes?

Response:

Thank you for your suggestion. We performed a control analysis evaluating the CNV amplitude and CNV slope recorded over the posterior and frontal electrodes.

Concerning the **posterior cluster** (pooled electrodes Pz, P1, P2, POz, PO3 and PO4), no effect of the CNV amplitude ($p > .480$) nor the CNV slope ($p > .264$) was reported (see Figure below).

Concerning the **anterior cluster** (pooled electrodes FCz, AFz, F1 and F2), no effect of the CNV amplitude ($p > .172$) nor the CNV slope ($p > .245$) was reported (see Figure below).

Given the lack of statistical effect, and the fact that our selection of electrodes was based on the literature, we only mention the lack of statistical significance in the method section: "It can be noted that no significant effect was observed when selecting a cluster over posterior or frontal electrodes (data not shown)." (lines 417-419).

Figure 6A and b: oscillation plots should be shown separately for each group (NVGPs and VGPS)

Response:

Figure 6A and B now display data for each group separately.

Methods

Participants: The VGP and NVGP group are not gender matched, and probably also not age matched. This needs to be controlled.

Response:

Thank you for your suggestion. We performed t-tests and χ^2 -tests, and actually did not find a significant difference in gender or age between VGPs and NVGPs. This information is now reported in the manuscript.

"T-tests revealed no difference in age ($p = .312$), ..." (line 331).

Line 587: How many trials in each block?

Were practice trials included?

Response:

Thank you for pointing out this missing information. It has been added to the manuscript.

"Participants performed four blocks of 120 trials for each task" (line 358).

"The experiment started with 25 training trials." (line 359).

Methods: how many trials are included in each group? For saccadic reaction times as well as ERP responses?

Response:

Thank you for asking for more precision. We added the following information to the manuscript.

Concerning the saccadic data, "Further analysis was performed on 455 and 450 remaining trials in average in VGPs and NVGPs, respectively" (line 384).

Concerning the ERP data, "The procedure rejected a mean average of 36 trials ($SD = 7$) and 35 trials ($SD = 8$) in VGPs and NVGPs, respectively." (line 410).

Figure 1 B: I think it would be helpful if the timeline is mentioned with regards to the two possible scenarios: short and long time period for the warning signal, that is two separate figures. This would

help to follow the segmentation procedure for the CNV. For instance, the predictive cues are not included in the time line, but this would help to understand the segmentation.

Thank you for your remark. The cues are represented in green in the figure, and they are predictive in the temporal condition. Separate timelines for the two foreperiods and two cueing conditions have been added in Figure 1B.

Also, the authors talk about foreperiod – could more conventional terms such as interstimulus interval or SOA be used here?

Response:

Indeed, ITIs and SOAs are very conventional terms. However, in tasks exploring the ability to orient attention in time, or to implicitly benefit from the passage of time, the conventional term found in the literature is Foreperiod. Therefore, we mentioned the relatedness in the Introduction but prefer to keep the term Foreperiod in the remaining of the manuscript. We do not mention SOAs, since the interval started with the offset of the initial cue rather than its onset.

“Participants were aware of the two possible foreperiods (also called inter-stimulus intervals, ITIs), i.e. 400 ms (short FP) or 1000 ms (long FP).” (lines 68).

Line 167: This time interval starts at the time of the earliest possible occurrence of the target and ends when the target appears.

Does it mean when the next target appears?

Response:

Thank you for your question, as the quoted sentence indeed needs clarification. The answer is no. The target can occur at T1 (400 ms) or T2 (1000 ms). In the analysis referred to, we analyzed trials with the target appearing at T2 (1000 ms) only, and the ‘earliest possible occurrence of the target’ simply meant 400 ms after the cue offset. To avoid confusion, we added a clarification in the sentence before.

“A two-way rANOVA with the factors Group and Cue was conducted on the magnitude of the centro-parietal CNV (Figure 4) recorded within the 400-1000 ms time interval in trials with long FP only (i.e. when the target appeared at 1000 ms).” (lines 182-184). We also specified again the delays (400 and 1000 ms) in the following sentence.

Results:

Line 87: VGPs are believed to be impulsive -here another study shows evidences that VGPs are not impulsive

Dye, M. W., Green, C. S., & Bavelier, D. (2009). Increasing speed of processing with action video games. *Current directions in psychological science*, 18(6), 321-326.

Response:

Thank you for the reference. We added the reference in the manuscript.

“VGPs are believed to be impulsive (but see 43) amongst the general public”. (line 94).

Abstract: Conclusion “may help remediate timing alterations in psychiatric populations” needs further clarification as the experiment and also the population is unrelated to any psychiatric disorder

Response:

Thank you for the comment, we agree this point required a clarification. The starting point of this study is a line of research showing that implicit processing of time, evaluated with the variable foreperiod task (as here), appears impaired in patients with schizophrenia (for a review, see <https://www.nature.com/articles/s44159-022-00038-y>).

Since our argument on the use of games in psychiatric populations can hardly be developed in the abstract, we now clarified this point in the Introduction:

“We need to know which types of mechanisms are enhanced in video game players, to better understand the impact of video game, and how they may help pathological groups. For example, implicit temporal mechanisms appear to be impaired in patients with schizophrenia (see 24 for a review), and video games have been proposed as a potential rehabilitation tool for psychiatric disorders such as schizophrenia ²⁵. Knowing how video game play shapes brain mechanisms and behaviors will help to adapt these rehabilitation tools to pathologies”. (lines 54-58)

Introduction: In the introduction the hypothesis of “learning to learn” and enhanced “attentional control functions” should be introduced as underlying principles why gamers outperform non-gamers in many

different tasks. For instance, the principle of “learning to learn” might allow gamers to learn the perceptual templates faster than in video-game players (see also Bejjanki et al., 2014; Zhang et al., 2021). Enhanced attentional control functions and its neural correlates have been discussed to be one underlying mechanism of enhanced perceptual and attentional control functions.

Bavelier, D., Achtman, R. L., Mani, M., & Föcker, J. (2012). Neural bases of selective attention in action video game players. *Vision research*, 61, 132-143.

Bejjanki, V. R., Zhang, R., Li, R., Pouget, A., Green, C. S., Lu, Z. L., & Bavelier, D. (2014). Action video game play facilitates the development of better perceptual templates. *Proceedings of the National Academy of Sciences*, 111(47), 16961-16966.

Föcker, J., Mortazavi, M., Khoe, W., Hillyard, S. A., & Bavelier, D. (2019). Neural correlates of enhanced visual attentional control in action video game players: an event-related potential study. *Journal of Cognitive Neuroscience*, 31(3), 377-389.

Green, C. S., Pouget, A., & Bavelier, D. (2010). Improved probabilistic inference as a general learning mechanism with action video games. *Current biology*, 20(17), 1573-1579.

Zhang, R. Y., Chopin, A., Shibata, K., Lu, Z. L., Jaeggi, S. M., Buschkuhl, M., ... & Bavelier, D. (2021). Action video game play facilitates “learning to learn”. *Communications biology*, 4(1), 1-10.

Response:

Thank you for all the references. We integrated your suggestion in the introduction.

“This enhancement involves increased ability to learn on the fly⁴⁻⁶ and improved attentional control^{7,8}” (lines 38-39).

The introduction gives the impression that a training study has been run – but this is not the case. We need to know which type of mechanisms are enhanced through gaming – as this is a cross-sectional study, causal conclusions cannot be drawn and the study also does not provide a unique answer to this question. Training studies are needed in order to answer the causal relationship here.

Response:

You are absolutely right. We clarify this point across the article and specifically stated that the article presents a cross-sectional study in the Introduction.

“In this cross-sectional study, we investigated...” (line 59).

Line 57: “whether reaction time performance” – what about other dependent variables such as accuracy or dprime?

Response:

Thank you again for the comment. As mentioned in the **Response** above:

Accuracy has been analyzed as anticipation errors (lines 94-96 in Manuscript; lines 3-14 in Supplemental Results). We reported the main effects of the Group (less errors in VGPs than NVGPs), the Cue (less errors in temporal than neutral cueing conditions) and the Foreperiod (less errors in short than long FP).

However, d' values could not be calculated in such a task. Our task is very easy and induce a very low percentage of errors.

Line 49: “However, except for one investigation, these studies explored spatial but not temporal aspects of attention”. Please also add: Green, C. S., & Bavelier, D. (2003). Action video game modifies visual selective attention. *Nature*, 423(6939), 534-537. The attentional blink task as an example about “attention over time” has been used here as well.

Response:

Thank you for your suggestion. The text has been adapted and the reference added.

“A large number of studies revealed that playing action video games improves explicit spatial attention mechanisms, their evolution over time and development, or explicit time discrimination abilities 8,16–23”. (lines 51-53)

Reviewers' comments:

Reviewer #1 (Remarks to the Author):

I appreciate the reviewer's responsiveness to the previous round of review. In all cases, they've made revisions to the manuscript that addressed my previous comments/concerns.

Reviewer #2 (Remarks to the Author):

The authors addressed all my previous comments in a satisfactory fashion. I congratulate them for this nice work.

Reviewer #3 (Remarks to the Author):

The authors have thoroughly revised the manuscript and provided additional details and analysis.

I still have two questions:

For Figure 3C the authors have mentioned that they verified the correlation would not be driven by outliers but there is no explanation about the procedure (at least I cannot see it). Could this be further explained (e.g. excluding $N = x$ outliers (specify number of gamers / non-gamers) analysis reveals a similar result pattern)? Additionally, I recommend adding a footnote. Furthermore, it is not clear whether the statistics reported below are including all participants (with outliers) or are already corrected (without outliers), it would be good to add the sample size here for each group.

Figure 3C: separate correlations in each group should be documented-is the correlation driven by outliers?

"Response: Thank you for your important remark. We verified that the correlation was indeed not driven by outliers. The correlation was specific to the VGP group. We corrected the text in the manuscript and the Figure 3C. "Interestingly, a Pearson correlation revealed a significant negative correlation between the latency of the peak inhibition and the benefit from the passage of time ($r = -.46$, $p = .028$) in VGPs but not in NVGPs ($r = -.04$, $p = .861$), independently of the cueing condition." (lines 168-170)"

The second question is if it might be possible that video-game players have more VR experiences than non-video-game players and are thus more familiar with the settings than non-video game players which might impact the task performance (e.g. less cybersickness etc). Did the authors control for this and ask their participants about possible VR experiences?

Response to Reviewers

Article: COMMSBIO-22-0592

Reviewer #3 (Remarks to the Author):

The authors have thoroughly revised the manuscript and provided additional details and analysis.

Dear reviewer, thank you again for helping us to improve our work in clarifying our research method.

I still have two questions:

For Figure 3C the authors have mentioned that they verified the correlation would not be driven by outliers but there is no explanation about the procedure (at least I cannot see it). Could this be further explained (e.g. excluding $N = x$ outliers (specify number of gamers / non-gamers) analysis reveals a similar result pattern)?

Additionally, I recommend adding a footnote. Furthermore, it is not clear whether the statistics reported below are including all participants (with outliers) or are already corrected (without outliers), it would be good to add the sample size here for each group.

Previous Question and Response:

Figure 3C: separate correlations in each group should be documented-is the correlation driven by outliers?

"Response: Thank you for your important remark. We verified that the correlation was indeed not driven by outliers. The correlation was specific to the VGP group. We corrected the text in the manuscript and the Figure 3C. "Interestingly, a Pearson correlation revealed a significant negative correlation between the latency of the peak inhibition and the benefit from the passage of time ($r = -.46$, $p = .028$) in VGPs but not in NVGPs ($r = -.04$, $p = .861$), independently of the cueing condition." (lines 168-170)"

Response:

Thank you for your additional questions. Indeed, we did not provide further details because our analysis did not highlight the existence of outliers in Figure 3C. Yet, we agree that the reader will be interested in having the disclosure about the absence of outliers.

To detect potential outliers, the presence of data points outside the interval of ± 3 SD from the mean was verified. The procedure reported no outliers within the VGPs ($N=23$) nor the NVGPs ($N=23$) group.

The sample sizes used for the correlation analysis have been added in the manuscript (lines 169 & 201) and these details are now reported in a footnote page 3:

"¹ All data points were within the interval of ± 3 SD from the mean. Thus, no outliers were detected. All participants were included in the correlation analysis. "

The second question is if it might be possible that video-game players have more VR experiences than non-video-game players and are thus more familiar with the settings than non-video game players which might impact the task performance (e.g. less cybersickness etc). Did the authors control for this and ask their participants about possible VR experiences?

Response:

Thank you for your remark. Indeed, the exposure to VR setups was discussed with the participants. Unfortunately, we did not quantify this exposure to VR. Therefore, it is hard to tell anything about the potential impact of VR exposure on the performance to the task.

Cybersickness was indeed a crucial factor for this experiment. First, pre-tests verified that the present VR setup was not inducing cybersickness. Second, cybersickness was discussed with each participant before and during testing, but not quantified through a questionnaire.

Yet, no participants reported experiencing cybersickness during or after the experiment, nor related symptoms (e.g. headache, stomach ache, dizziness). This remains an interesting information.

We have now added the following lines in the Method section:

“The exposure to VR experience was not quantified nor compared between the two groups. No participant reported any feeling of cybersickness during or after the experiment.” (lines 336-338)

Also, the lack of quantification of the VR expose has been commented in the Discussion section:

“Also, future research should consider the possible influence of prior experience of VR, while keeping in mind that VR interventions could represent an affordable and engaging remediation tool for time perception alterations in psychiatric populations 25.” (lines 320-322)